



# Non-equilibrium interplay between gas-particle partitioning and multiphase chemical reactions of semi-volatile compounds: mechanistic insights and practical implications for atmospheric modeling of PAHs

Jake Wilson[1], Ulrich Pöschl[1], Manabu Shiraiwa[2,*], and Thomas Berkemeier[1,*]

[1]Multiphase Chemistry Department, Max Planck Institute for Chemistry, Mainz, Germany
[2]Department of Chemistry, University of California, Irvine, CA, USA

**Correspondence:** Thomas Berkemeier (t.berkemeier@mpic.de) and Manabu Shiraiwa (m.shiraiwa@uci.edu)

**Abstract.** Polycyclic aromatic hydrocarbons (PAHs) are carcinogenic air pollutants. The dispersion of PAHs in the atmosphere is influenced by gas-particle partitioning and chemical loss. These processes are closely interlinked and may occur at vastly differing timescales, which complicates their mathematical description in chemical transport models. Here, we use a kinetic model that explicitly resolves mass transport and chemical reactions in the gas and particle phases to describe and explore the dynamic and non-equilibrium interplay of gas-particle partitioning and chemical losses of PAHs on soot particles. We define the equilibration timescale $\tau_{eq}$ of gas-particle partitioning as the $e$-folding time for relaxation of the system to the partitioning equilibrium. We find this metric to span seconds to hours depending on temperature, particle surface area and the type of PAH. The equilibration time can be approximated using a time-independent equation $\tau_{eq} \approx \frac{1}{k_{des}+k_{ads}}$, which depends on the desorption rate coefficient $k_{des}$ and adsorption rate coefficient $k_{ads}$, both of which can be calculated from experimentally-accessible parameters. The model reveals two regimes in which different physical processes control the equilibration timescale: a *desorption-controlled* and an *adsorption-controlled* regime. In a case study with the PAH pyrene, we illustrate how chemical loss can perturb the equilibrium particulate fraction at typical atmospheric concentrations of $O_3$ and OH. For the surface reaction with $O_3$, the perturbation is significant and increases with the gas-phase concentration of $O_3$. Conversely, perturbations are smaller for reaction with the OH radical, which reacts with PAHs on both the surface of particles and in the gas phase. Global and regional chemical transport models typically approximate gas-particle partitioning with instantaneous equilibration approaches. We highlight scenarios in which these approximations deviate from the explicit-coupled treatment of gas-particle partitioning and chemistry presented in this study. We find that the discrepancy between solutions depends on the operator-splitting time step and the choice of time step can help to minimize the discrepancy. The findings and techniques presented in this work are not only relevant for PAHs, but can also be applied to other semi-volatile substances that undergo chemical reactions and mass transport between the gas and particle phase.





## 1 Introduction

Polycyclic aromatic hydrocarbons (PAHs) are air pollutants that are structurally characterized by their fused aromatic ring systems (Keyte et al., 2013). Given their carcinogenic properties (Boström et al., 2002), developmental toxicity (Billiard et al., 2008) and abundance in the environment (Ravindra et al., 2008), PAHs pose a risk to human health (Kim et al., 2013).

PAHs are semi-volatile compounds that may exist in the gas phase, adsorbed on the surface of aerosol particles, or absorbed into the bulk of particles. The transport and distribution between these phases is referred to as gas-particle partitioning. For PAHs, an accurate model description of gas-particle partitioning is needed to interpret monitoring data, determine atmospheric burden and lifetime, and ultimately assess the hazards their emissions pose to human health.

In equilibrium, the flux of a semi-volatile species from the gas phase to the particle surface is equal to the flux that desorbs back into the gas phase. The state of the system at equilibrium can be described mathematically with thermodynamic entities (Junge, 1977; Yamasaki et al., 1982; Pankow, 1994). The equilibrium gas-particle partitioning of semi-volatile compounds is determined by both adsorption onto the particle surfaces and absorption into the particle bulk. The relative contributions of these processes depend on the concentrations, composition and phase state of particles. Physicochemical properties of a compound, such as the octanol-air partition coefficient $K_{OA}$ (Finizio et al., 1997; Harner and Bidleman, 1998), the soot-air partition coefficient $K_{SA}$ (Dachs and Eisenreich, 2000; Lohmann and Lammel, 2004) and Abraham descriptors (Arp et al., 2008; Shahpoury et al., 2016) are typically used to predict the position of equilibrium. In terms of surface adsorption, soot or black carbon particles may be especially relevant for the gas-particle partitioning of PAHs as they exhibit large energies of desorption (Kubicki, 2006; Guilloteau et al., 2010) and are often co-emitted (Miguel et al., 1998).

Field observations of PAH gas-particle partitioning in the form of particulate fractions often differ from the predictions of equilibrium models (Lohmann et al., 2000; Mandalakis et al., 2002; Terzi and Samara, 2004; Callén et al., 2008; Akyüz and Çabuk, 2010; Lammel et al., 2010; Wei et al., 2015). These discrepancies can be induced by perturbations to the equilibrium such as emissions, changes in temperature, dry and wet deposition, or chemical reaction (Pankow and Bidleman, 1992), but only persist if the rate of the adsorption-desorption kinetics is slow compared to rate of the perturbing process. In other words, partitioning equilibrium will still be maintained if the equilibration timescale is shorter than the perturbation timescale. This means that accurate knowledge of equilibration timescales is vital to assess whether non-equilibrium effects may occur.

Gas-particle partitioning is an important process in chemical transport models (CTMs) which describe the long-distance transport and chemical degradation of atmospheric constituents such as PAHs (Shrivastava et al., 2017; Mu et al., 2018). In CTMs, PAHs are partitioned, transformed and transported in discrete time steps, often using the method of operator splitting. With operator splitting, the partitioning equilibrium is restored at each model time step through instantaneous equilibration (Galarneau et al., 2014) rather than treating gas-particle partitioning continuously. PAH concentrations predicted by CTMs have been shown to depend on the employed treatment of gas-particle partitioning. For instance, Lammel et al. (2009) found that using different equilibrium partitioning models influenced atmospheric cycling, the total environmental fate, and long-range transport potential of PAHs. Friedman et al. (2014) found that implementing a partitioning scheme in which PAHs slowly evaporate from aerosol particles yielded the better agreement between observed and simulated concentration and par-



titioning data compared to the instantaneous equilibration approach. Overall, CTMs that assume equilibrium partitioning tend to be more common than those accounting for mass-transfer limitations explicitly, as can be seen from a recent review of partitioning methods in regional-scale transport models of SOA (McFiggans et al., 2015).

Equilibration timescales of gas-partitioning may be estimated theoretically, using analytical equations or numerical models. By solving analytical transport equations, the equilibration timescales of partitioning for volatile inorganic compounds were found to depend on the size of particles (Meng and Seinfeld, 1996). More recently, there have been numerical simulations for secondary organic aerosol (SOA) as a function of temperature and relative humidity (Shiraiwa and Seinfeld, 2012; Li and Shiraiwa, 2019). Alternatively, equilibration timescales may also be obtained experimentally. For example, Saleh et al. (2013) found the equilibration timescale of SOA formed by $\alpha$-pinene ozonolysis to be less than 30 min following a perturbation in temperature. Furthermore, the interplay between partitioning and multiphase reaction of OH with alkanes was shown to influence the distribution of product isomers (Zhang et al., 2015).

For PAHs, several studies have investigated the timescales of gas-particle partitioning from the perspective of absorptive partitioning. Rounds and Pankow used a radial diffusion model to investigate the kinetic limitations of partitioning resulting from diffusion of a semi-volatile compound absorbed within a particle (Rounds and Pankow, 1990). Odum et al. (1994) additionally included a parameter to accounting for mass-transfer limitations at the surface. In chamber experiments, Kamens et al. (1995) examined the equilibration timescales of PAHs. However, an in-depth analysis of the important case of PAH adsorption onto the surface of soot remains elusive. In recent years, the desorption rate coefficients of PAHs from soot have been experimentally parameterized over a range of atmospherically-relevant temperatures (Guilloteau et al., 2010). However, a systematic comparison between the equilibration timescales of partitioning and the timescales of loss processes has not been carried out.

In this study, we examine the timescales of gas-particle partitioning and chemical loss of PAHs with a kinetic model in which both processes are explicitly coupled. The model uses the conventions of the PRA framework (Pöschl et al., 2007) and is based on the kinetic double-layer model for aerosol surface chemistry (K2-SURF; Shiraiwa et al., 2009). We quantify the equilibration timescales of six PAHs on the model surface of solid soot particles for different temperatures and particle number concentrations (Sect. 3.2). We illustrate how the combination of slow partitioning and chemical loss of PAHs can perturb the particulate fraction from equilibrium (Sect. 3.3.2) and alter chemical lifetime (Sect. 3.3.1). We apply the knowledge gained from the kinetic model calculations to the description of gas-particle partitioning in CTMs by comparing the explicitly-coupled solution to a method mimicking operator splitting with instantaneous equilibration and evaluate the performance of both methods in different scenarios.

## 2 Methods

### 2.1 Kinetic Model

A modified version of the kinetic double-layer model K2-SURF is used for all simulations (Shiraiwa et al., 2009). The original K2-SURF model consists of a near-surface gas phase and surface layer, with gas diffusion from the far-surface gas phase represented by a correction factor (Fig. A1). In this study, we added explicit treatment of gas diffusion to K2-SURF to track





gas-phase PAH concentrations. PAHs reversibly desorb and adsorb between the aerosol particle surface and the near-surface

gas phase. The rate coefficient for PAH desorption from the particle surface $k_{\mathrm{des}} = Ae^{-E_A/RT}$ (s$^{-1}$) depends on temperature $T$ and two parameters determined from experiment: the Arrhenius factor $A$ and the activation energy of desorption $E_A$ (Guilloteau et al., 2010). $R$ is the gas constant. Aerosol particles are assumed to be monodisperse and to consist of a spherical, impenetrable solid carbon core. The system is closed with respect to aerosol particles and PAH in all simulations. In simulations involving chemical reactions, the system is open with respect to oxidants, i.e. gas-phase OH and O$_3$ concentrations are fixed.

The differential equations in Eqs. 1-3 describe the time evolution of $[\mathrm{PAH}]_{\mathrm{g}}$ (cm$^{-3}$), $[\mathrm{PAH}]_{\mathrm{gs}}$ (cm$^{-3}$) and $[\mathrm{PAH}]_{\mathrm{s}}$ (cm$^{-2}$), which are the concentrations of PAH in the gas phase, near-surface gas phase and on the surface of aerosol particles, respectively. $J_{\mathrm{des}}$ (cm$^{-2}$ s$^{-1}$), $J_{\mathrm{ads}}$ (cm$^{-2}$ s$^{-1}$) and $J_{\mathrm{diff}}$ (s$^{-1}$) are the desorption flux, adsorption flux and gas diffusion flux. Each flux term is described in detail in section A of the appendix.

$$\frac{\mathrm{d}[\mathrm{PAH}]_{\mathrm{g}}}{\mathrm{d}t} = -J_{\mathrm{diff}}N_{\mathrm{p}} - L_{\mathrm{g}} \tag{1}$$

$$\frac{\mathrm{d}[\mathrm{PAH}]_{\mathrm{gs}}}{\mathrm{d}t} = \frac{(J_{\mathrm{des}} - J_{\mathrm{ads}})d_{\mathrm{p}}^2\pi + J_{\mathrm{diff}}}{V_{\mathrm{gs}}} \tag{2}$$

$$\frac{\mathrm{d}[\mathrm{PAH}]_{\mathrm{s}}}{\mathrm{d}t} = -J_{\mathrm{des}} + J_{\mathrm{ads}} - L_{\mathrm{s}} \tag{3}$$

The surface area of a single aerosol particle with diameter $d_{\mathrm{p}}$ (cm) is $d_{\mathrm{p}}^2\pi$, $V_{\mathrm{gs}}$ (cm$^3$) is the volume of gas in the near-surface gas phase for a single aerosol particle and $N_{\mathrm{p}}$ (cm$^{-3}$) is the particle number concentration. $L_{\mathrm{g}}$ (cm$^{-3}$ s$^{-1}$) is the rate of chemical loss in the gas phase and $L_{\mathrm{p}}$ (cm$^{-2}$ s$^{-1}$) is the rate of chemical loss in the particle phase. Reactions of PAHs within

the near-surface gas phase are considered to be negligible due to the small fraction of PAHs in this volume. Sources of PAHs are not considered in this study.

## 2.2 Chemical reactions

The surface reaction between PAH and O$_3$ is modeled using a Langmuir-Hinshelwood mechanism, including reversible adsorption of O$_3$ onto the surface of aerosol particles and reaction of surface-adsorbed O$_3$ with surface-adsorbed PAH.

$$\mathrm{O}_{3(\mathrm{gs})} \rightleftharpoons \mathrm{O}_{3(\mathrm{s})} \tag{R1}$$

$$\mathrm{PAH}_{(\mathrm{s})} + \mathrm{O}_{3(\mathrm{s})} \rightarrow \mathrm{product}_{(\mathrm{s})} \tag{R2}$$




The rate coefficient for reaction of PAH with $O_3$, $k_{\text{PAH+O3}} = 2.7 \times 10^{-17}\,\text{cm}^2\,\text{s}^{-1}$, and the corresponding mass-transport parameters are taken from Shiraiwa et al. (2009) (Table A1). Reaction products are treated as inert and non-volatile. Note that

the reaction between $O_3$ and benzo(a)pyrene on the surface of soot has been suggested to involve the formation of long-lived reactive oxygen intermediates (ROI; Shiraiwa et al., 2011). Such a detailed chemical mechanism is beyond the scope of this study, which instead focuses on the interaction of partitioning and chemistry, and has thus been omitted for simplicity. The gas-phase reaction between $O_3$ and PAH is considered negligible and is therefore not included (Keyte et al., 2013). The reaction between PAH and OH is accounted for in both the gas phase and on the surface of particles.

$$\text{PAH}_{(g)} + \text{OH}_{(g)} \rightarrow \text{product}_{(g)} \tag{R3}$$

$$\text{PAH}_{(s)} + \text{OH}_{(gs)} \rightarrow \text{product}_{(s)} \tag{R4}$$

The gas-phase reaction between PAH and OH is modeled with the rate coefficient $k_{\text{PAH+OH}} = 6.58 \times 10^{-11}\,\text{cm}^3\,\text{s}^{-1}$ (theoretical calculation for pyrene at 298 K, Zhang et al., 2014). We do not consider temperature dependence of chemical rate coefficients in this model. The reaction between PAH and OH on the surface of particles is treated with an Eley-Rideal like

mechanism using a surface reaction probability of 0.32 (obtained for pyrene, Bertram et al., 2001) and assuming an OH gas diffusion coefficient $D_g$ of $0.21\,\text{cm}^2\,\text{s}^{-1}$ (Tang et al., 2014). The uptake of OH onto the surface of particles is considered to be irreversible.

### 2.3 Particulate fraction

The measured distribution of PAHs (and other semi-volatiles) between the particle phase and the gas phase is commonly

described with the particulate fraction $\Phi$, i.e. the fraction of total PAHs associated with aerosol particles (Eq. 4).

$$\Phi = \frac{[\text{PAH}]_p}{[\text{PAH}]_p + [\text{PAH}]_g} \tag{4}$$

The total concentration of PAH adsorbed on the surface of aerosol particles $[\text{PAH}]_p$ $(\text{cm}^{-3})$ is the product of the surface area of a single particle $d_p^2\pi$ with diameter $d_p$, the particle number concentration $N_p$, and surface concentration of PAH, $[\text{PAH}]_s$ $(\text{cm}^{-2}$; Eq. 5).

$$[\text{PAH}]_p = d_p^2\pi N_p[\text{PAH}]_s \tag{5}$$

### 2.4 Equilibration timescale $\tau_{\text{eq}}$

To quantify the time for PAHs to reach their equilibrium distribution between the gas phase and the particle phase we use the equilibration timescale ($\tau_{\text{eq}}$), defined as e-folding time for the relaxation of the system to gas-particle partitioning equilibrium.





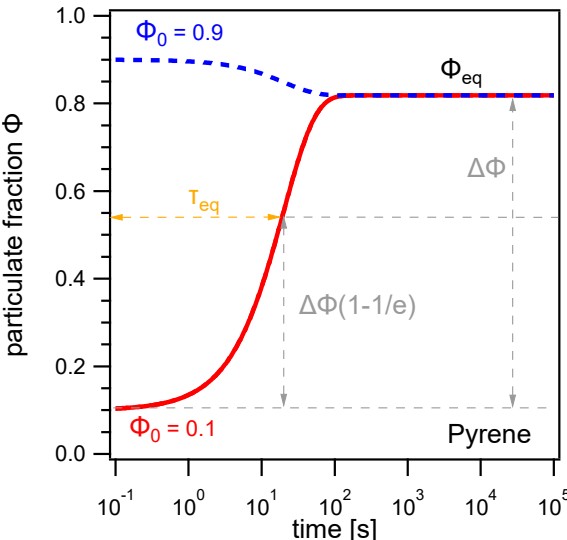

**Figure 1.** The evolution of the particulate fraction $\Phi$ of pyrene with respect to time, starting from initial particulate fractions $\Phi_0 = 0.1$ (red) and $\Phi_0 = 0.9$ (blue). The equilibration timescale $\tau_{\mathrm{eq}}$ is defined as the time required for the system to achieve 63.2 % of $\Delta\Phi$, the difference between $\Phi_0$ and $\Phi_{\mathrm{eq}}$.

Figure 1 shows results from a kinetic box model simulation with a concentration of pyrene in air of $5 \times 10^5$ cm$^{-3}$, temperature
$T = 298$ K, particle number concentration $N_{\mathrm{p}} = 1 \times 10^4$ particles cm$^{-3}$, and particle diameter $d_{\mathrm{p}} = 200$ nm. No chemical loss of pyrene is included here. $\tau_{\mathrm{eq}}$ is obtained numerically from model outputs by interpolating the time required by the system to achieve $1 - \frac{1}{e}$ (i.e. $\approx 63.2$ %) of the difference $\Delta\Phi$ between an initial particulate fraction $\Phi_0$ and the equilibrium particulate fraction $\Phi_{\mathrm{eq}}$.

In this example, pyrene reaches $\Phi_{\mathrm{eq}}$ after $\approx 2$ minutes and the equilibration timescale is independent of the initial particulate
fraction, i.e. $\tau_{\mathrm{eq}}$ is the same regardless of whether $\Phi_0 = 0.1$ or $\Phi_0 = 0.9$. In fact, $\tau_{\mathrm{eq}}$ is found to be independent of the choice of $\Phi_0$ for most conditions due to the first-order and hence mono-exponential nature of the adsorption and desorption processes. This allows for consistent intercomparison across different temperatures and particle number concentrations without changing starting distributions. Exceptions may occur in cases where surface adsorption is not strictly a first-order process, either due to surface saturation effects or gas phase diffusion limitations. These conditions occur at very low particle number concentrations
(typically $< 1 \times 10^3$ particles cm$^{-3}$) and further details are given in Appendix B.

## 3 Results and discussion

### 3.1 Extreme cases of multiphase chemistry and partitioning interaction

Three extreme cases can be formulated when partitioning and chemical-loss processes of a semi-volatile compound take place at different relative timescales (Fig. 2).





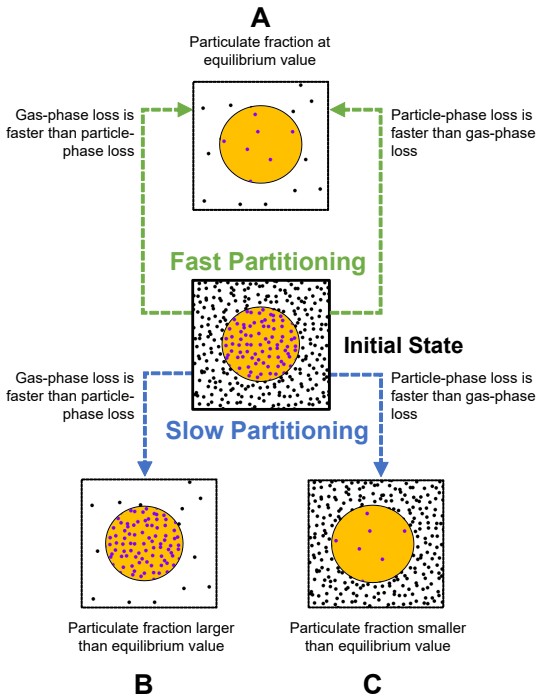

**Figure 2.** Schematic on how the gas-particle partitioning equilibrium of a semi-volatile compound may be perturbed from an initial state (center) due to chemical loss, depending on equilibration timescales. If the timescales of partitioning are shorter than the timescales of chemical loss, the system is able to maintain equilibrium (A). However, the combination of rapid gas-phase loss and slow replenishment from the particle phase increases the particulate fraction above the equilibrium value (B). In turn, the combination of rapid particle-phase loss and slow replenishment by condensation decreases particulate fraction below the equilibrium value (C).

When the timescale of partitioning is short compared to the timescales of chemical loss, molecules are redistributed quickly between both phases (case A, Fig. 2). In this case, the relative amounts of gas- and particle-phase species will remain very close to their equilibrium values ($\Phi \approx \Phi_{eq}$). This is independent of whether molecules are lost primarily from the gas phase or from the particle phase.

In contrast, if the timescale of the partitioning process is slow and the chemical loss rates from the gas and particle phase differ substantially, the particulate fraction will be perturbed from its equilibrium value (cases B and C, Fig. 2). When the loss rate in the gas phase exceeds the loss rate in the particle phase, the particulate fraction increases beyond its equilibrium value ($\Phi > \Phi_{eq}$; case B, Fig. 2). However, when the loss rate in the particle phase is greater than that in the gas phase, the particulate fraction decreases ($\Phi < \Phi_{eq}$; case C, Fig. 2).

Unlike these scenarios, chemical loss and partitioning timescales may not differ substantially. Likewise, chemical losses are likely to take place in both phases simultaneously. Every real system must therefore be seen as superposition of these cases. The extent to which perturbation occurs depends upon the difference between partitioning and chemical reactions timescales. An in-depth discussion on the magnitude of perturbation is provided in Sect. 3.3.



Hence, two preconditions are required for the particulate fraction $\Phi$ of the system to be perturbed from the equilibrium particulate fraction predicted by equilibrium partitioning theory $\Phi_{eq}$: 1) Slow partitioning relative to the timescale of chemical loss and 2) an imbalance of chemical loss between the gas and particle phases.

If timescales of chemical loss and partitioning were known for all natural systems, they could be classified and mathematically treated in the respective limiting case. In this manuscript, we: 1) estimate the partitioning timescales of PAH on soot as a function of atmospheric conditions, 2) compare these timescales to typical chemical loss rates in order to investigate whether perturbations from equilibrium exist, and 3) explore the implications of treating partitioning and chemistry separately in chemical transport models.

### 3.2   Partitioning equilibration timescales for PAHs on soot

$\tau_{eq}$ depends on the molecular structure of the PAH, particle number concentration and temperature. This is explored in the following section with a series of simulations using a fixed total concentration of PAHs in air of $5 \times 10^5$ cm$^{-3}$ and particles with a diameter of 50 nm.

#### 3.2.1   PAH molecular structure

#### 3.2.2   Particle number concentration

The effect of varying the particle number concentration $N_p$ on the equilibration timescale shows a distinct behavior (Fig. 3a): $\tau_{eq}$ is particle number-independent at lower $N_p$, while $\tau_{eq}$ is particle number-dependent at higher $N_p$. The equilibration timescales of the less strongly adsorbed PAHs including anthracene, fluoranthene and pyrene are not significantly affected by particle number concentration until a fairly high threshold particle number concentration is achieved ($\approx 10^5$, $10^4$ and $10^4$ particles cm$^{-3}$, respectively).

Once the threshold particle number concentration is reached, a linear relationship in the double logarithmic dependence of equilibration timescale and particle number concentration emerges. The more strongly adsorbed PAHs, chrysene, benzo(e)pyrene and benzo(a)pyrene, reach this limit at a much lower $N_p$ ($\approx 10^2$ particles cm$^{-3}$). This can be understood when looking at Fig. 3b, in which the equilibration timescale of pyrene is shown together with the individual timescales of desorption $\tau_{des}$ (gray dashed line, calculated with Eq. 6) and adsorption $\tau_{ads}$ (gray dotted line, calculated with Eq. 7).

$$\tau_{des} = \frac{1}{k_{des}} = A e^{-E_A / RT} \tag{6}$$

$$\tau_{ads} = \frac{1}{k_{ads}} = \frac{1}{(1 - \theta_s)\alpha_{s,0} d_p^2 \pi N_p \omega / 4} \approx \frac{1}{\alpha_{s,0} d_p^2 \pi N_p \omega / 4} \tag{7}$$

In the limit of an adsorbate-free surface, adsorption and desorption are first-order processes with respect to the near-surface gas and surface concentrations of PAH respectively and can therefore be described with rate coefficients $k_{ads}$ (s$^{-1}$) and $k_{des}$



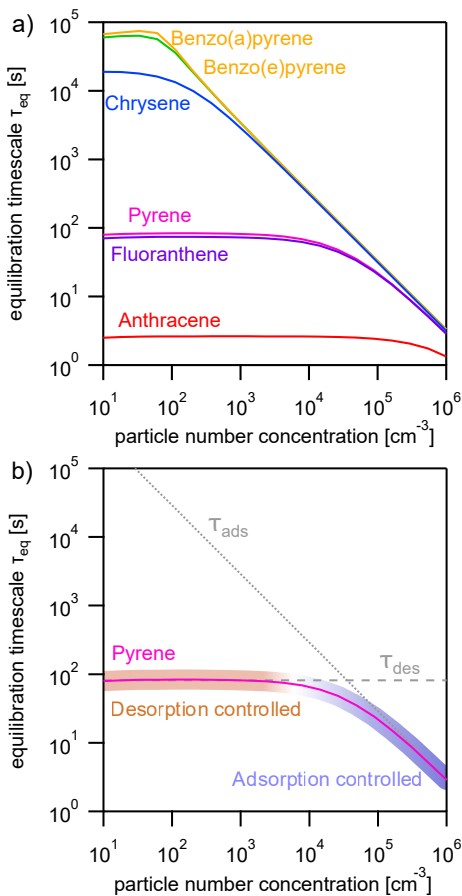

**Figure 3.** a) Equilibration timescale $\tau_{\text{eq}}$ for different PAHs as a function of particle number concentration at a constant temperature ($T = 298$ K). b) Comparison of the equilibration timescale of pyrene to the timescales of adsorption $\tau_{\text{ads}}$ and desorption $\tau_{\text{des}}$, highlighting the transition between adsorption-controlled and desorption-controlled behavior.

(s$^{-1}$). The desorption timescale $\tau_{\text{des}}$ depends on the Arrhenius factor $A$ and the activation energy for desorption from the aerosol particle surface ($E_{\text{A}}$), the gas constant $R$ and temperature $T$. The adsorption timescale $\tau_{\text{ads}}$ depends on the surface accommodation coefficients on an adsorbate-free substrate $\alpha_{\text{s},0}$, the particle number concentration $N_{\text{p}}$, the surface area of a single aerosol particle $d_{\text{p}}^2\pi$ with diameter $d_{\text{p}}$ and the mean thermal velocity $\omega$. The surface coverage $\theta_{\text{s}}$ is very small for typical

particle number concentrations and will therefore be neglected in the following. In general, $\tau_{\text{eq}}$ can be approximated as a function of both timescales according to Eq. 8 (see appendix for details of the terms and derivation). This approximation holds as long as gas diffusion is sufficiently fast and does not limit equilibration.

$$\tau_{\text{eq}} \approx \frac{1}{k_{\text{des}} + k_{\text{ads}}} = \frac{1}{Ae^{-E_{\text{A}}/RT} + \alpha_{\text{s},0}d_{\text{p}}^2\pi N_{\text{p}}\omega/4} \tag{8}$$



If one process (desorption or adsorption) dominates the behavior of $\tau_{\mathrm{eq}}$, the system can be said to fall into an *adsorption-controlled* regime (highlighted for pyrene with red shading) or a *desorption-controlled* regime (highlighted with blue shading). A multi-step process in which mass is lost and transferred in one direction can be described analagously to a series of resistors in an electrical circuit and the term *limiting* can be used to describe the slowest step. In contrast, the gas-particle partitioning system is a reversible system in which mass is transferred in both directions and the relative rates of these mass-transfer processes determine the position of equilibrium. We therefore observe in Fig. 3b (and also Fig. 4b) that the equilibration time is determined primarily by the fastest process (i.e. that with the shortest timescale). We therefore adopt the term *controlled* to characterize this behavior.

In the low particle number concentration limit, the system is in a *desorption-controlled* regime and the equilibration timescale is thus strongly influenced by strength of the PAH-soot interaction, which explains the large differences in equilibration timescale between PAHs in Fig. 3a. In the high particle number concentration limit, the equilibration timescale is determined primarily by the adsorption of PAH onto particles from the near-surface gas phase and is therefore independent of PAH type as can be seen from the convergence of curves in Fig. 3a. The equilibration timescale here coincides with the adsorption timescale $\tau_{\mathrm{ads}}$ and the system is in an *adsorption-controlled* regime. The transition between both regimes occurs where $\tau_{\mathrm{ads}}$ intersects $\tau_{\mathrm{des}}$ and coincides with the point $\Phi_{\mathrm{eq}} = 0.5$. At this specific point, equal amounts of PAH are in the gas and particle phases and the timescales of desorption and adsorption contribute equally to the equilibration time.

As surface coverages $\theta_{\mathrm{s}}$ are very small and PAHs generally have surface accommodation coefficients on an adsorbate-free substrate of $\alpha_{\mathrm{s,0}} = 1$ (Julin et al., 2014), we find in this study a special case of the *adsorption-controlled* regime where molecular collision of gas molecules is the sole controller of partitioning. For adsorbates with lower $\alpha_{\mathrm{s,0}}$, the adsorption timescale would be longer and the system may be in the *desorption-controlled* regime.

### 3.2.3 Temperature

The effect of varying temperature $T$ on the equilibration timescale shows a behavior similar to the one seen for the particle number concentration (Fig. 4a): $\tau_{\mathrm{eq}}$ is temperature-independent at low $T$, while $\tau_{\mathrm{eq}}$ is temperature-dependent and begins to decrease at higher $T$. For the most weakly adsorbed PAH anthracene, $\tau_{\mathrm{eq}}$ begins decreasing at 240 K towards higher $T$ and at 298 K is already less than 5 s. The equilibration timescales for fluoranthene and pyrene begin decreasing at $\approx 260$ K and at 298 K are both less than 100 s. Strongly adsorbed PAHs including chrysene, benzo(e)pyrene and benzo(a)pyrene do not undergo a significant change in equilibration timescale in the investigated temperature range.

Again, the *adsorption-controlled* and *desorption-controlled* regimes explain this behavior (Fig. 4b). Between 210 K and 240 K, PAH molecules possess little kinetic energy and are prevented from escaping into the gas phase, thus exhibiting long desorption lifetimes (Fig. A3) and high equilibrium particulate fractions (Fig. A2b). As most PAH is adsorbed on the surface of aerosol particles, molecular collision determines equilibration time. The system is in the *adsorption-controlled* regime, highlighted for pyrene with blue shading and signified by the coincidence with the adsorption timescale $\tau_{\mathrm{ads}}$ (gray dotted line). The number of collisions between gas-phase PAHs and particles slightly increases as the thermal velocity increases, but this effect is much smaller compared to the effect of temperature increase on desorption rates. Note that the surface





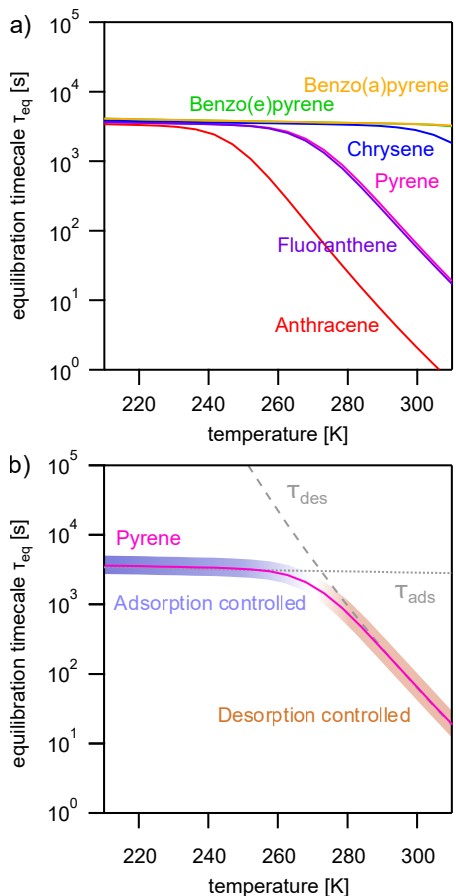

**Figure 4.** a) Equilibration timescale $\tau_{eq}$ for different PAHs as a function of temperature at a constant particle number concentration ($N_p = 1 \times 10^3 \ \text{particles cm}^{-3}$). b) Comparison of the equilibration timescale of pyrene to the timescales of adsorption $\tau_{ads}$ and desorption $\tau_{des}$, highlighting the transition between *adsorption-controlled* and *desorption-controlled* behavior.

accommodation coefficient is assumed to be temperature-independent in this study. Overall, upon increase in temperature, the desorption process becomes increasingly important. At high temperature, the system is in the *desorption-controlled* regime,

highlighted for pyrene with red shading in Fig. 4b and signified by the coincidence with the timescale of desorption $\tau_{des}$ (gray dashed line).

## 3.3 Interplay of multiphase chemistry and partitioning

Chemical reactions with $O_3$ and OH are important loss processes for PAHs. If the rate of chemical loss is fast relative to gas-particle partitioning, the gas-particle distribution may be perturbed from its equilibrium state (cf. Fig. 2, cases B and C).

This effect is exemplified for pyrene by including surface chemistry with $O_3$ (0, 1, 10 and 100 ppb) or gas-phase and surface chemistry with OH (0, 0.01, 0.1 and 1 ppt) in the model. 10 ppb is representative of surface background $O_3$ concentrations



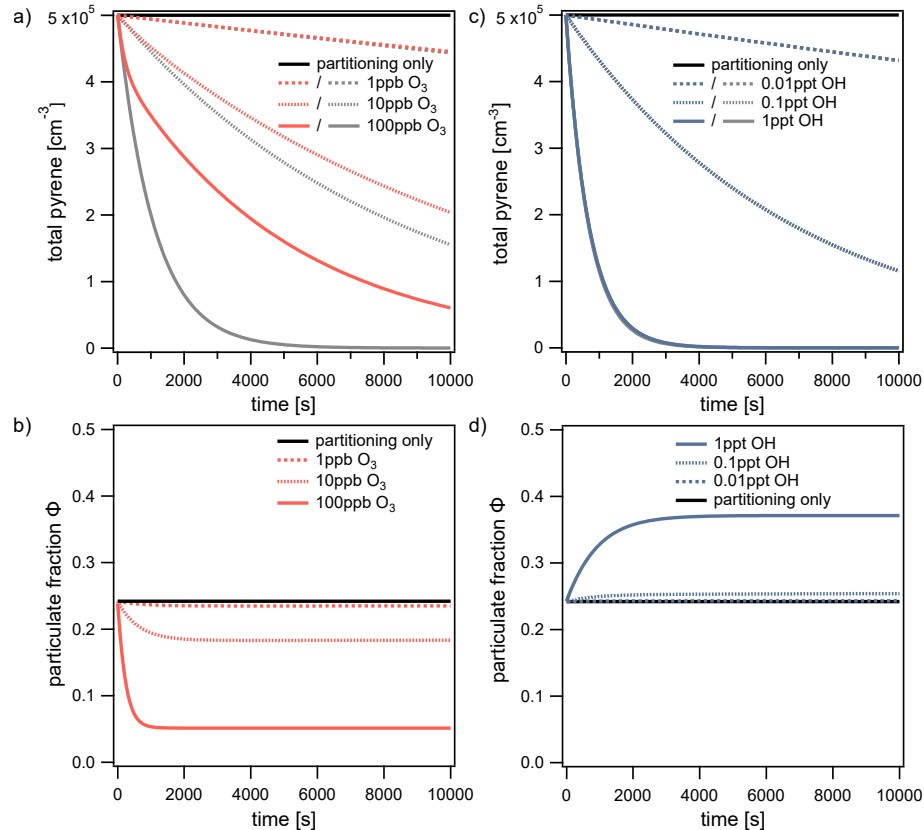

**Figure 5.** The total pyrene concentration (gas phase + particle phase) with respect to time at different concentrations of a) $O_3$ and b) OH. $O_3$ reacts only with pyrene on the surface of particles, whereas OH reacts with pyrene in simultaneous gas and surface reactions. The total concentration of pyrene determined in a fictional scenario with infinitely fast partitioning is shown with gray lines. For the same simulations, the particulate fraction of pyrene with respect to time is shown: c) $O_3$ and d) OH. The equilibrium particulate fraction $\Phi_{eq}$ is shown with a black line.

(Vingarzan, 2004), while 100 ppb $O_3$ is characteristic of concentrations at more polluted sites (Wang et al., 2017). An OH concentration of 0.01 ppt is representative of concentrations measured at night (Geyer et al., 2003), while 0.1 ppt is representative of daytime concentrations (Stone et al., 2012) and 1 ppt is an upper limit only encountered in highly polluted conditions

(Hofzumahaus et al., 2009) and smoke plumes (Hobbs et al., 2003). We employ the following conditions in the pyrene-soot system: $T = 280$ K, $N_p = 1 \times 10^3$ particles cm$^{-3}$, $d_p = 50$ nm. At the start of the simulation, the initial total concentration of pyrene ($5 \times 10^5$ cm$^{-3}$) is distributed between the gas and particle phases according to the particulate fraction expected at equilibrium (i.e. $\Phi_0 = \Phi_{eq} = 0.24$). The kinetic rate coefficients for all reactions are given in Sect. 2.2. Oxidant concentrations are fixed during simulations.





### 3.3.1 Non-equilibrium effects on chemical lifetime and particulate fraction

Figure 5a shows the decrease in the total (gas + particle) concentration of pyrene at different $O_3$ concentrations. The chemical lifetime of pyrene is determined by calculating the time needed for the concentration to reach $1/e$ (i.e. $\approx 36.8\,\%$) of its initial value. As the concentration of $O_3$ increases from 1 ppb to 10 ppb and 100 ppb, the chemical lifetime of pyrene subsequently decreases to 23.9 h, 3.1 h and 1.2 h, respectively. It is informative to compare these chemical lifetimes to those calculated by assuming partitioning is infinitely fast (i.e. the particulate fraction is fixed to $\Phi_{eq}$ during simulations). In this fictional scenario, the lifetimes of pyrene are significantly shorter at 23.1 h, 2.4 h and 0.3 h and correspond to decreases of 3 %, 23 % and 75 %, respectively. This comparison demonstrates that partitioning and chemical loss are closely interlinked. In this specific case, slow partitioning prolongs the lifetime of pyrene.

This effect can be understood by observing the change in particulate fraction over time during each of the simulations (Fig. 5b). As each simulation proceeds, the particulate fraction $\Phi$ drops below the equilibrium particulate fraction $\Phi_{eq}$ and eventually reaches a quasi-steady state $\Phi_{qs}$. At $O_3$ concentrations of 10 ppb and 100 ppb, the particulate fractions reach values of $\Phi_{qs} = 0.18$ and $0.05$, respectively. This effect can be explained by slow partitioning: chemical loss reduces the surface concentration of pyrene faster than its replenishment from the gas phase (non-equilibrium case C in Fig. 2). In the quasi-steady state, chemical loss and repartitioning are balanced. Importantly, both values differ significantly from $\Phi_{eq}$. In contrast, when the $O_3$ concentration is low enough (1 ppb) the particulate fraction remains approximately equal to its value at equilibrium ($\Phi \approx \Phi_{eq} = 0.24$). At this $O_3$ concentration, the rate of partitioning is sufficiently high so that pyrene lost from the particle surface can be fully replenished from the gas phase (equilibrium case A in Fig. 2). Hence, non-equilibrium behavior increases with oxidant concentration.

Figure 5c shows the decrease in total concentration of pyrene due to the simultaneous gas and surface reactions with OH. The lifetimes of pyrene with OH concentrations of 0.01 to 0.1 and 1 ppt, are 18.9 h, 1.9 h and 0.2 h, respectively. Nearly identical lifetimes are obtained if partitioning is assumed to be infinitely fast, thus indicating that non-equilibrium effects on chemical lifetime are insignificant for this system. Figure 5d shows that in contrast to the behavior of the $O_3$ system, the highest concentration of OH perturbs the particulate fraction to a quasi-steady state above its equilibrium value. The particulate fraction reaches a quasi-steady state with a value of $\Phi_{qs} = 0.37$ at 1 ppt OH. Although chemical loss takes in both phases simultaneously, the turnover of pyrene is higher in the gas phase. The particulate fraction thus increases, characteristic of the non-equilibrium case B in Figure 2. At lower concentrations of OH, the extent of the perturbation becomes only slight ($\Phi_{qs} = 0.25$ at 0.1 ppt) and eventually disappears (0.01 ppt in Fig. 5d). Hence, non-equilibrium effects on particulate fraction can be significant, even if they were insignificant for chemical lifetime. This is due to the short reaction timescale of the OH-pyrene system compared to its partitioning timescale: pyrene reaches $1/e$ of its initial concentration before the quasi-steady state is established.

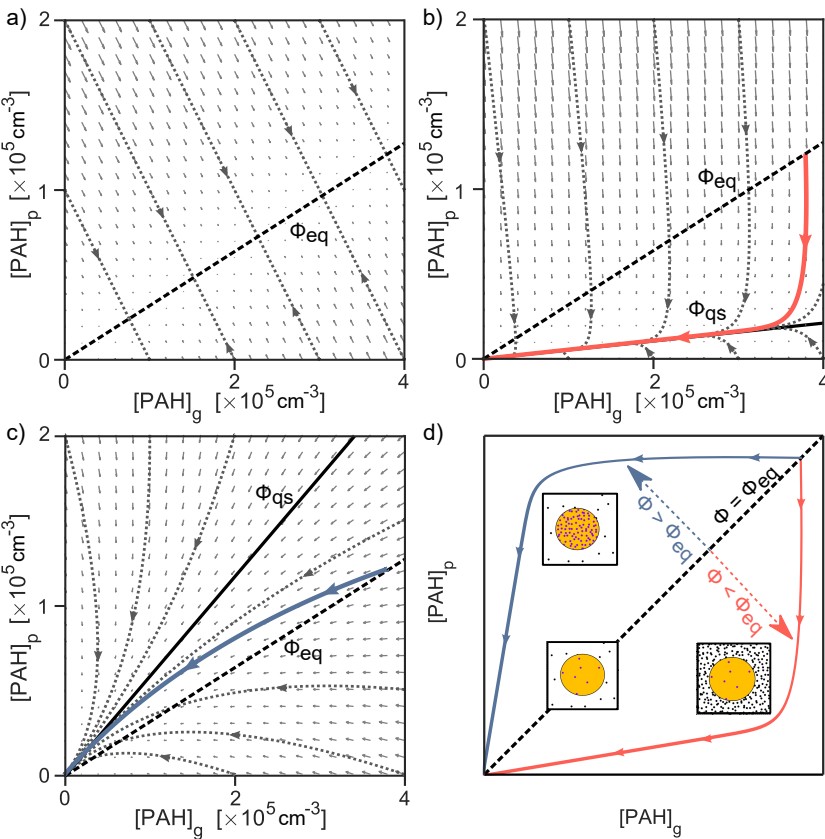

**Figure 6.** The interplay of partitioning and chemistry illustrated as simulated trajectories in the phase space of gas- and particle-phase concentrations, $[PAH]_g$ and $[PAH]_p$, of pyrene. a) partitioning only, b) partitioning and chemical loss from the particle phase due to surface reaction with $O_3$ (100 ppb, red solid line), c) partitioning and chemical loss from the gas and particle phase due to reaction with OH (1 ppt, blue solid line), and d) comparison to schematics in Fig. 2. The direction of each small gray vector arrow indicates the extent to which pyrene is being lost or transferred between phases, and its length is proportional to the rate of change. The nullcline representing equilibrium particulate fraction in panel a) ($\Phi_{eq} = 0.24$, black dashed line) is shown in panels b) and c), alongside the respective slow manifold representing the quasi-steady-state particulate fraction ($\Phi_{qs}$, solid black line).

### 3.3.2 Visualization of non-equilibrium effects with phase portraits

The dynamic behavior of the system may be visualized as a trajectory in the phase space of gas-phase and particle-phase pyrene concentrations, $[PAH]_g$ and $[PAH]_p$ (Fig. 6). At every point in the phase portrait, a vector illustrates how the system would change with time. Here, the direction of each vector arrow indicates the extent to which pyrene is being lost or trans-
ferred between phases, and its length indicates the rate of change. The exact characteristics of the phase portrait depend on temperature, available particle surface area, the strength of the PAH-soot interaction and the rate of the chemical reactions





involved. For a system where pyrene partitions without chemical loss, all trajectories converge onto a central line at which the system stops changing, known in mathematics as the nullcline (Fig. 6a). This line represents the point of gas-particle partitioning equilibrium. The slope of the line represents the dimensionless gas-particle partitioning coefficient $K_\mathrm{p}$, from which the

equilibrium particulate fraction $\Phi_\mathrm{eq}$ can be obtained (Eq. 9).

$$[\mathrm{PAH}]_\mathrm{p} = K_\mathrm{p}[\mathrm{PAH}]_\mathrm{g} = \frac{\Phi_\mathrm{eq}}{1 - \Phi_\mathrm{eq}}[\mathrm{PAH}]_\mathrm{g} \tag{9}$$

As seen previously, chemical reactions may cause perturbation of the partitioning equilibrium. Such a perturbation would be indicated by deviation of trajectories from the nullcline in the phase portrait. The difference between perturbed and equilibrium system is depicted for $[\mathrm{O}_3] = 100$ ppb in Fig. 6b. The vector field fundamentally changes and the trajectory of an

exemplary simulated system (red solid line) does not converge to the nullcline obtained in Fig. 6a, despite starting at equilibrium conditions $\Phi_\mathrm{eq}$ (shown as black dashed line for reference). Instead, the trajectory converges onto a central trajectory termed the slow manifold (Fraser, 1988). All trajectories in this system (represented with gray dotted lines) converge towards this manifold, irrespective of initial conditions. After approaching the slow manifold, the trajectory proceeds towards the origin (i.e. full depletion of pyrene) with a constant slope. This constant slope indicates that a constant quasi-steady-state particulate

fraction $\Phi_\mathrm{qs} = 0.05$ has been reached. The deviation of the nullcline (Fig. 6a) and the slow manifold (Fig. 6b) can be used to indicate the extent of non-equilibrium effects in a multiphase chemical reaction system. For example, Fig. 6c shows that for the reaction with 1 ppt OH, the discrepancy between the simulation trajectory and the partitioning nullcline is much smaller due to simultaneous loss in the gas and particle phases. The slow manifold here runs above the partitioning nullcline and is reached only just before all pyrene is consumed (compare solid blue lines in Fig. 5). Fig. 6d shows how the nullcline and the

slow manifolds above or below it can be interpreted using the diagrams in Fig. 2.

### 3.4   Implications for chemical transport models (CTMs)

In the previous sections, an explicit, coupled model of partitioning and chemistry is used. This means that mass-transport and chemical-loss processes are simultaneously evaluated in a set of differential equations. Hereon, this is referred to as the explicit-coupled approach (EC). As the explicit-coupled approach is computationally expensive, CTMs often treat the partitioning and

chemical loss of PAHs separately using operator splitting (Brasseur and Jacob, 2017). Instantaneous equilibration (IE) is one type of operator-splitting approach: at each model time step ($\Delta t$), the gas-particle distribution of PAH is reset to the partitioning equilibrium (estimated by temperature, particle number concentration, PAH, and particle type) and chemical loss is then further integrated separately, starting from the newly established equilibrium. Time steps of global and regional transport models used to study PAH are typically around $15$ min (Galarneau et al., 2014) or $30$ min (Sehili and Lammel, 2007).

### 3.4.1   Influence of model time step length

The solution obtained using the IE approach can differ from the EC solution. Using the surface reaction of $\mathrm{O}_3$ ($100$ ppb) with pyrene in the particle phase, we demonstrate that the magnitude and sign of this difference varies between $\Delta t = 4$ min,



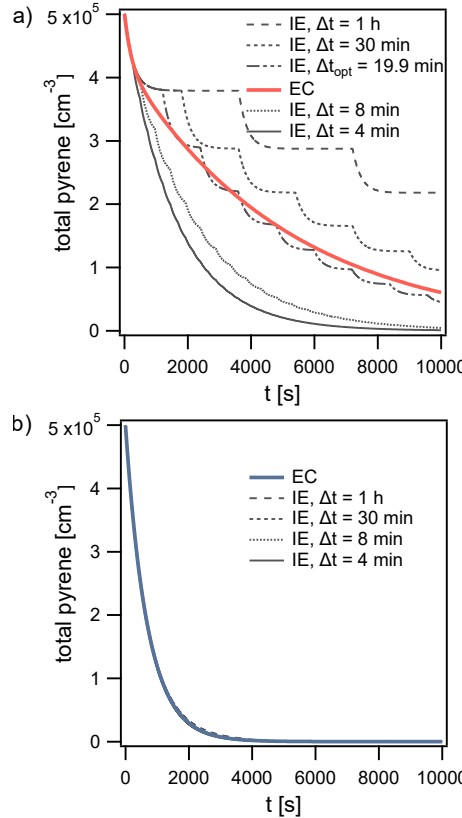

**Figure 7.** Time evolution of total pyrene concentration in model calculations, comparing the explicit-coupled (EC) to the operator splitting with instantaneous equilibration (IE) solution ($\Delta t = 4$ min, $8$ min, $30$ min and $1$ h) for systems with a) surface reaction of particle-phase pyrene with $100$ ppb $O_3$ and b) reaction of gas- and surface-bound pyrene with $1$ ppt OH. The operator-splitting time step leading to the smallest deviation is depicted as $\Delta t_{opt}$.

$8$ min, $30$ min and $1$ h (Fig. 7a). The following conditions are used in the simulations: $T = 280$ K, $d_p = 50$ nm, $N_p = 1 \times 10^3$ particles cm$^{-3}$.

The lifetime of pyrene is underestimated when using time steps $\Delta t$ of $4$ and $8$ min (Fig. 7a), but overestimated with $\Delta t$ of $30$ min and $1$ h. In this example, an optimal time step exists, for which the deviation from EC is minimized, $\Delta t_{opt} = 19.9$ min (Fig. 7a). It is close to the equilibration timescale of gas-particle partitioning $\tau_{eq}$ of pyrene, which is around $15$ min under these conditions. $\tau_{eq}$ could thus serve as initial guess for $\Delta t_{opt}$. $\Delta t_{opt}$ is determined using a golden-section search optimization algorithm (Kiefer, 1953) to minimize the absolute difference between EC and IE model outputs. Of note, a deviation between

IE and EC and hence a dependence of the model result on the operator-splitting time step only arises if a significant departure from partitioning equilibrium occurs. Under equilibrium partitioning conditions, a range of sufficiently short $\Delta t$ can describe the system accurately. In an example with OH ($1$ ppt) reacting with gas-phase and surface-bound pyrene, all IE calculations





produce negligible errors, irrespective of the time step used (Fig. 7b). This is due to the particulate fraction being very close to $\Phi_{eq}$ until the majority of PAH has reacted (cf. Fig. 5d).

### 335    3.4.2    Deviation from explicit-coupled (EC) approach

The discrepancy between the EC and IE solutions not only depends on the length of $\Delta t$, but also on the relative rates of partitioning and chemical loss. The discrepancy can be quantified with an error metric, $E_{loss}$, which can be interpreted as the relative difference in loss rates (Eq. 10). $\Delta[PAH]_{EC}(t)$ and $\Delta[PAH]_{IE}(t)$ are the accumulated losses of PAH at each time point $t$ using EC and IE, respectively (Eqs. 11 and 12). This metric is chosen as it detects discrepancies in model solutions independent

of the absolute turnover, which is important when comparing scenarios at high and low oxidant concentrations. $E_{loss}$ ranges between -1 and 1, and is evaluated until either $t_{90\%}$ or $t = 24$ h is reached. $E_{loss}$ is positive when the IE solution overpredicts the loss of pyrene compared to the reference EC solution, and negative when loss is underestimated.

$$E_{loss} = \frac{1}{n} \sum_{t=1}^{n} \frac{\Delta[PAH]_{IE}(t) - \Delta[PAH]_{EC}(t)}{\Delta[PAH]_{EC}(t) + \Delta[PAH]_{IE}(t)} \tag{10}$$

$$\Delta[PAH]_{EC}(t) = [PAH]_{EC}(0) - [PAH]_{EC}(t) \tag{11}$$

$$\Delta[PAH]_{IE}(t) = [PAH]_{IE}(0) - [PAH]_{IE}(t) \tag{12}$$

Figure 8 shows the extent and direction of deviation of IE from EC in a case study of PAH surface chemistry in which the desorption rate coefficient $k_{des}$ is varied between $5\times10^{-8}$ and $5 \times 10^{-1}$ s$^{-1}$, and the concentration of O$_3$ between 0 and 120 ppb for IE time steps of $\Delta t = 1$ min and $\Delta t = 30$ min. When $\Delta t = 1$ min (Fig. 8a), IE overestimates PAH loss compared to EC, indicated by red coloring. Deviation is largest when $k_{des}$ is between $1\times10^{-4}$ and $1\times10^{-2}$ s$^{-1}$. Here, the IE time step

of 1 min causes PAH to transfer onto particles at an artificially high rate. This increases the particle-phase concentration of PAH and results in faster chemical loss. The IE solution hence shows less non-equilibrium effects of slow partitioning on multiphase chemistry compared to the reference EC model. At the highest $k_{des}$ ($> 1 \times 10^{-2}$ s$^{-1}$), non-equilibrium effects of slow partitioning still occur, but in the *desorption-controlled* regime (cf. Fig. 4 at 280 K) an increase in $k_{des}$ leads to a reduction in equilibration timescale. This not only leads to weaker non-equilibrium effects of slow partitioning in the EC solution, but also

to a better match between EC equilibration timescale and IE time step. Hence, the discrepancy between IE and EC approach is reduced, as evident by the more faint red coloring. At low $k_{des}$ ($< 1\times10^{-5}$ s$^{-1}$), most PAH is located on the surface of particles at all times and re-partitioning of gas-phase PAH after depletion of particle-phase PAH is negligible. Thus, no deviation of the IE from the EC approach occurs.

In contrast, with a time step of $\Delta t = 30$ min (Fig. 8b), the IE approach generally underestimates the loss of PAH compared

to the EC approach, indicated by blue coloring. The largest underestimations are found at high $k_{des}$ and high $[O_3]_g$. Underestimation of PAH loss occurs because the re-partitioning induced by the longer IE time step of 30 min is slower than the true



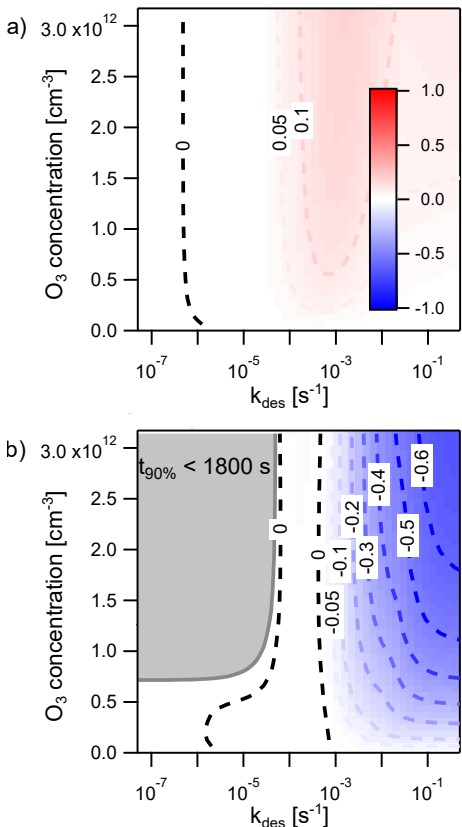

**Figure 8.** Discrepancy between the explicit-coupled solution EC and the operator-splitting solution IE for surface reaction of particle-phase pyrene with $O_3$ (100 ppb), determined with the error metric $E_{\mathrm{loss}}$. Regions where IE overestimates the loss of PAH are indicated in red (0 $< E_{\mathrm{loss}} \leq 1$) and where it underestimates PAH loss in blue ($-1 \geq E_{\mathrm{loss}} < 0$). The result is shown using two different operator-splitting time step lengths a) $\Delta t = 1$ min and b) $\Delta t = 30$ min. The dark gray area indicates the conditions under which the time taken for total pyrene to deplete by 90 % ($t_{90\%}$) is less than a single operator-splitting time step.

equilibration rate in the EC model. In this scenario, the IE approach thus leads to stronger non-equilibrium effects of slow partitioning compared to the EC model. When the equilibration timescale becomes shorter, at high $k_{\mathrm{des}}$ ($> 1 \times 10^{-2}$ s$^{-1}$), the discrepancy between the IE and the EC solution further increases, especially at high $[O_3]_g$. Notably, at $k_{\mathrm{des}} \approx 1 \times 10^{-4}$ s$^{-1}$, the

IE approach slightly overestimates loss of PAH at high $[O_3]_g$. This is due to the EC equilibration timescale dropping below the equivalent of $\Delta t = 30$ min just before non-equilibrium effects of slow partitioning vanish at the lowest $k_{\mathrm{des}}$. In between, a zero contour (labeled '0') and hence no deviation between both methods occurs when $k_{\mathrm{des}} \approx 1 \times 10^{-3}$ s$^{-1}$. Here, the IE approach matches the EC equilibration timescale by alternatingly underestimating and overestimating the concentration of PAH at different points of the simulation and a cancellation of errors occurs (cf. Fig. 7a). This case is distinct from the left zero contour

at $k_{\mathrm{des}} \approx 5 \times 10^{-5}$ s$^{-1}$, in which departure from equilibrium does not occur and both methods return truly identical results (compare Figs. A5a and A5c). Another region exists: for $k_{\mathrm{des}} < 1 \times 10^{-4}$ s$^{-1}$, the simulation proceeds for less than 30 min





and therefore less than one IE time step is evaluated (Fig. A5). In this region partitioning effectively does not take place and we chose not to report the numerical value of $E_{\text{loss}}$.

$E_{\text{loss}}$ is negligibly small when the concentration of $[\text{OH}]_g$ is varied between 0 and 0.5 ppt with reaction in both the gas phase and on the surface of particles (Fig. A4a). Unlike the example with $O_3$, PAH loss rates due to reaction with OH in each phase are similar enough that $\Phi$ is not perturbed far from $\Phi_{\text{eq}}$ under these conditions. Often, in chemical transport models only the gas-phase reaction of PAH with OH is included. Varying the concentration of OH between 0 and 0.5 ppt with reaction in the gas phase only causes IE to overestimate the loss of PAH (red area, Fig. A4b).

## 4 Atmospheric implications

This study shows that the chemical loss of polycyclic aromatic hydrocarbons (PAHs) and their partitioning between the gas and particle phases are closely interlinked. The equilibration timescales of adsorptive partitioning are quantified for six PAHs on the surface of soot. Our model predicts that equilibration timescales range from seconds to hours depending on temperature, available particle surface area and molecular structure of the PAH. We highlight the molecular processes governing this timescale with two regimes: *adsorption-controlled* and *desorption-controlled* partitioning.

Soot constitutes only a fraction of total ambient aerosol particles (Pöschl, 2005). Thus, a logical next step will be to investigate how equilibration timescales vary for other types of particle surfaces. For example, given the weaker desorption energies of PAHs such as anthracene on the surface of NaCl ($75.3\,\text{kJ}\,\text{mol}^{-1}$) compared to soot ($87.9\,\text{kJ}\,\text{mol}^{-1}$; Chu et al., 2010), one would expect equilibration timescales on NaCl to be shorter.

On other particle types, PAHs molecules can undergo absorptive partitioning by diffusing through surface layers into the bulk of the particle. For secondary organic aerosol (SOA), the particle phase state influences the rates of condensation and evaporation (Shiraiwa and Seinfeld, 2012). Equilibration timescales for PAHs are therefore also expected to be dependent on particle composition and humidity. Depending on the controlling regime, it might be expected that absorption lengthens the equilibration timescale compared to adsorption on a solid surface. The complex interplay of partitioning and reaction in the gas and particle phases plays a critical role in the growth of SOA particles (Shiraiwa et al., 2013; Berkemeier et al., in review, 2020) and departure from partitioning equilibrium adds to this complexity (Cappa and Wilson, 2011). However, the role of bulk diffusion in determining equilibration timescales is beyond the scope of this study and will be investigated in a follow-up study that builds on the framework provided here.

Chemical reaction of pyrene with $O_3$ on the surface of particles perturbs the particulate fraction from partitioning equilibrium at atmospherically-relevant oxidant concentrations. As the extent of this perturbation increases with concentration of $O_3$, the largest deviations from equilibrium particulate fraction are likely to take place in the most polluted air. The reaction of pyrene with OH in both phases results in much smaller perturbations. In general, the biggest deviations from equilibrium particulate fraction are expected for low-volatility PAHs when atmospheric conditions induce slow partitioning (i.e. cold temperatures and low particle number concentrations). Other chemical-loss processes may also be important for PAHs such as the reaction with $NO_3$ in both the gas phase (Zhang et al., 2014) and the particle phase (Gross and Bertram, 2008), as well as aqueous-phase





photodegradation processes (Fasnacht and Blough, 2002). These reactions must eventually be studied simultaneously in order to establish whether loss in both phases balances out or whether perturbation from equilibrium takes place.

It should also be noted that while in this study simulations involving chemical loss are initialized at the point of equilibrium ($\Phi_0 = \Phi_{eq}$), in reality PAHs may be emitted in a state far from partitioning equilibrium. Depending on the prevailing loss processes, such an effect could both enhance or inhibit perturbations from equilibrium.

Non-chemical loss processes, such as dry and wet deposition remove PAHs from the gas and the particle phases at different rates and may also cause perturbations from partitioning equilibrium. The fastest loss processes, i.e. those operating at the shortest timescales, will cause the greatest perturbations. In the case of polybrominated diphenyl ethers (PBDEs), Li et al. (2015) attempted to include the effect of loss by deposition on partitioning and derived an equation for the partitioning coefficient assuming a steady state rather than equilibrium. However, given that for PAHs the estimated lifetimes due to dry

deposition (1 to 14 days) and wet deposition (5 to 15 months; Škrdlíková et al., 2011) are much longer than lifetimes due to chemical loss (in this study less than 24 h), chemical loss is expected to be the loss process that is most likely to perturb the partitioning equilibrium.

The methodology described in this study is universally applicable to semi-volatile compounds on solid surfaces if mass-transfer parameters and chemical reaction rate coefficients are available. In some cases it may be necessary to estimate these

parameters. In quantum mechanical calculations, graphene surfaces could be used as a model for soot and desorption energies. Such values are already available for PBDEs (Ding et al., 2014) and other small organic molecules (Lazar et al., 2013).

It has to be noted that an explicitly coupled solution of partitioning and chemical loss is computationally too expensive for inclusion in typical regional and global CTMs. Hence, alternative algorithms would be highly anticipated. Knowledge about the position of the partitioning steady state in the presence of chemical loss (as indicated by the slow manifold that can be

visualized in a phase portrait of gas and particle phase concentrations) could be used to develop such a method for global and regional models.





## Appendix A: Kinetic model

The flux of PAH molecules from the gas phase to the near-surface gas phase $J_{\mathrm{diff}}$ with concentrations $[\mathrm{PAH}]_{\mathrm{g}}$ and $[\mathrm{PAH}]_{\mathrm{gs}}$, respectively, is calculated with Eq. A1.

$$J_{\mathrm{diff}} = 2\pi(d_{\mathrm{p}} + 2\lambda)D_{\mathrm{g}}([\mathrm{PAH}]_{\mathrm{g}} - [\mathrm{PAH}]_{\mathrm{gs}}) \tag{A1}$$

The gas-phase diffusion coefficient $D_{\mathrm{g}}$ is fixed at $0.06\ \mathrm{cm}^2\,\mathrm{s}^{-1}$ for all PAH compounds, based on measurements for anthracene and pyrene in nitrogen (Siddiqi et al., 2009). $d_{\mathrm{p}}$ is the diameter of aerosol particles. The mean free path $\lambda$ is defined in Eq. A2.

$$\lambda = \frac{3D_{\mathrm{g}}}{\omega} \tag{A2}$$

The mean thermal velocity of a molecule $\omega$ depends on temperature $T$ and its molar mass $M$ (Eq. A3).

$$\omega = \sqrt{\frac{8RT}{\pi M}} \tag{A3}$$

The adsorption flux $J_{\mathrm{ads}}$ of molecules from the near-surface gas phase to the particle phase is described using Eq. A4.

$$J_{\mathrm{ads}} = \alpha_{\mathrm{s},0}(1 - \theta_{\mathrm{s}})J_{\mathrm{coll}} \tag{A4}$$

The surface accommodation coefficient on an adsorbate-free substrate $\alpha_{\mathrm{s},0}$ describes the probability that a molecule adsorbs 440 upon collision with an adsorbate-free aerosol particle and for PAH molecules is assumed to be $\alpha_{\mathrm{s},0} = 1$ (Julin et al., 2014). The total sorption layer coverage $\theta_{\mathrm{s}}$ is calculated as the sum of the fractional coverages of PAH and $\mathrm{O}_3$, $\theta_{\mathrm{PAH}}$ and $\theta_{\mathrm{O3}}$, respectively (Eq. A5). $\theta_{\mathrm{PAH}}$ and $\theta_{\mathrm{O3}}$ are calculated using the surface concentrations $[\mathrm{PAH}]_{\mathrm{s}}$ and $[\mathrm{O}_3]_{\mathrm{s}}$, and molecular cross sections $\sigma_{\mathrm{PAH}}$ and $\sigma_{\mathrm{O3}}$ of PAH and $\mathrm{O}_3$, respectively (Eq. A6). In order to estimate $\sigma$, each benzene-like ring of a PAH molecule is assumed to occupy $2 \times 10^{-15}\ \mathrm{cm}^2$. The collision flux $J_{\mathrm{coll}}$, i.e. the flux of molecules colliding with the surface, is defined in Eq. A7.

$$\theta_{\mathrm{s}} = \theta_{\mathrm{PAH}} + \theta_{\mathrm{O3}} \tag{A5}$$

$$\theta_{\mathrm{PAH}} = \sigma[\mathrm{PAH}]_{\mathrm{s}} \tag{A6}$$

$$J_{\mathrm{coll}} = \frac{[\mathrm{PAH}]_{\mathrm{gs}}\omega}{4} \tag{A7}$$





The temperature dependent desorption flux ($J_\text{des}$) due PAH molecules evaporating from the surface of an aerosol particle depends on the rate coefficient for desorption ($k_\text{des}$) and the surface concentration of PAH $[\text{PAH}]_\text{s}$ (Eq. A8).

$$J_\text{des} = k_\text{des}[\text{PAH}]_\text{s} \tag{A8}$$

$k_\text{des}$ depends on the Arrhenius factor ($A$) and the activation energy for desorption from the aerosol particle surface ($E_\text{A}$; Eq. A9).

$$k_\text{des} = Ae^{-E_\text{A}/RT} \tag{A9}$$

The temperature dependence of the desorption rate coefficient was previously determined for seven PAHs on fresh kerosene
soot (Guilloteau et al., 2008, 2010) and the obtained parameters are implemented in this model (see Table A1). These activation energies of desorption for PAHs on soot are consistent with those obtained theoretically on pure graphene (Lechner and Sax, 2014) and coronene (Kubicki, 2006). It should also be noted that different types of soots can have different effects on gas-particle partitioning (Mader and Pankow, 2002) and more aged soot may have a reduced affinity for PAH. Despite the simplifications of this model, we aim to provide a basis to which further complexity can be added.
Irreversible reactions between pyrene and either $O_3$ on the surface of aerosol particles or OH in the gas phase and on the surface of aerosol particles are investigated with the model. The equations of mass-transport for $O_3$ are identical to those for PAH and the corresponding parameters are reported in Tables A1 and A2. As the uptake of OH is considered to proceed via an Eley-Rideal mechanism, the diffusion of OH from the gas-phase to the near-surface gas phase is treated using a gas-phase diffusion correction factor $C_\text{g,OH}$ (Eq. A10). The full equation for $C_\text{g,OH}$ can be found in Eq. 14 of Pöschl et al. (2007).

$$[\text{OH}]_\text{gs} = C_\text{g,OH}[\text{OH}]_\text{g} \tag{A10}$$

The rate of PAH loss from the particle surface due to chemical reaction with OH $L_\text{s,OH}$ depends on probability $\gamma_\text{OH,PAH}$ that reaction occurs following collision of OH with PAH on the particle surface (Eq. A11). The rate of gas-phase PAH loss by OH $L_\text{g,OH}$ and the rate of PAH loss from the surface due to reaction with $O_3$ $L_\text{s,O3}$ are defined by Eq. A12 and Eq. A13, respectively. Further details of these reactions and their parameters can be found in section 2.2.

$$L_\text{s,OH} = \gamma_\text{OH,PAH}\theta_\text{PAH}J_\text{coll,OH} \tag{A11}$$

$$L_\text{g,OH} = k_\text{g}[\text{PAH}]_\text{g}[\text{OH}]_\text{g} \tag{A12}$$

$$L_\text{s,O3} = k_\text{s}[\text{PAH}]_\text{s}[\text{O}_3]_\text{s} \tag{A13}$$





## Appendix B: Derivation of equation for equilibration time

An approximate equation for equilibration time $\tau_{\text{eq}}$ (s) is obtained analytically using the relaxation time of a simple reversible
reaction (Bernasconi, 1976). The equation approximates the numerically-obtained results from the kinetic model and is derived
by assuming that gas-particle partitioning can be described as two first-order processes, adsorption and desorption, with rate
coefficients $k_{\text{des}}$ $(\text{s}^{-1})$ and $k_{\text{ads}}$ $(\text{s}^{-1})$. We find there is good agreement between the numerically-obtained results and the
approximate equation as long as the gas diffusion flux $J_{\text{diff}}$ does not significantly affect gas-particle partitioning (i.e. $[\text{PAH}]_{\text{g}} \approx$
$[\text{PAH}]_{\text{gs}}$) and surface crowding effects do not significantly inhibit adsorption of PAH onto the surface (i.e. $\theta_{\text{s}}$ is small). The
equilibration timescale of PAHs predicted by the equation are within $10\,\%$ of the numerically-obtained timescale among our
test conditions ($\geq 10^3$ particles $\text{cm}^{-3}$, $210\,\text{K} \geq T \geq 310\,\text{K}$). For a gas-phase $[\text{PAH}]_{\text{g}}$ $(\text{cm}^{-3})$ and particle-phase concentration
$[\text{PAH}]_{\text{p}}$ $(\text{cm}^{-3})$, the rate equation can be expressed in the following way

$$\frac{\text{d}[\text{PAH}]_{\text{g}}}{\text{d}t} = -\frac{\text{d}[\text{PAH}]_{\text{p}}}{\text{d}t} = k_{\text{des}}[\text{PAH}]_{\text{p}} - k_{\text{ads}}[\text{PAH}]_{\text{g}}. \tag{B1}$$

At equilibrium, the concentration of PAH in the gas and particle phases is $[\text{PAH}]_{\text{g,eq}}$ and $[\text{PAH}]_{\text{p,eq}}$. As there is no net transfer
in mass

$$\frac{\text{d}[\text{PAH}]_{\text{g}}}{\text{d}t} = k_{\text{des}}[\text{PAH}]_{\text{p,eq}} - k_{\text{ads}}[\text{PAH}]_{\text{g,eq}} = 0. \tag{B2}$$

The concentrations of PAH in the gas and particle phases can therefore be expressed as a displacement from their values at
equilibrium, $\Delta[\text{PAH}]_{\text{g}}$ and $\Delta[\text{PAH}]_{\text{p}}$ (Eq. B3 and Eq. B4).

$$[\text{PAH}]_{\text{g}} = [\text{PAH}]_{\text{g,eq}} + \Delta[\text{PAH}]_{\text{g}} \tag{B3}$$

$$[\text{PAH}]_{\text{p}} = [\text{PAH}]_{\text{p,eq}} + \Delta[\text{PAH}]_{\text{p}} \tag{B4}$$

Substitution of $\Delta[\text{PAH}]_{\text{g}} = -\Delta[\text{PAH}]_{\text{p}} = x$ into Eq. B3 and Eq. B4 allows the displacements from equilibrium to be expressed in terms of a single variable $x$ (Eq. B5 and Eq. B6).

$$[\text{PAH}]_{\text{g}} = [\text{PAH}]_{\text{g,eq}} + x \tag{B5}$$

$$[\text{PAH}]_{\text{p}} = [\text{PAH}]_{\text{p,eq}} - x \tag{B6}$$





Insertion of Eq. B5 and Eq. B6 into Eq. B2 gives

$$\frac{\mathrm{d}[\mathrm{PAH}]_\mathrm{g}}{\mathrm{d}t} = +k_\mathrm{des}([\mathrm{PAH}]_\mathrm{p,eq} - x) - k_\mathrm{ads}([\mathrm{PAH}]_\mathrm{g,eq} + x) = 0. \tag{B7}$$

Expansion of brackets, followed by insertion of Eq. B2 gives

$$\frac{\mathrm{d}[\mathrm{PAH}]_\mathrm{g}}{\mathrm{d}t} = -(k_\mathrm{des} + k_\mathrm{ads})x = 0. \tag{B8}$$

Insertion of Eq. B5 into Eq. B8 gives Eq. B9, then noting that $\mathrm{d}[\mathrm{PAH}]_\mathrm{g,eq}/\mathrm{d}t = 0$ gives Eq. B10.

$$\frac{\mathrm{d}([\mathrm{PAH}]_\mathrm{g,eq} + x)}{\mathrm{d}t} = \frac{\mathrm{d}[\mathrm{PAH}]_\mathrm{g,eq}}{\mathrm{d}t} + \frac{\mathrm{d}x}{\mathrm{d}t} = -(k_\mathrm{des} + k_\mathrm{ads})x = 0 \tag{B9}$$

$$\frac{\mathrm{d}x}{\mathrm{d}t} = -(k_\mathrm{des} + k_\mathrm{ads})x. \tag{B10}$$

Both sides are integrated (Eq. B11) and the equation rearranged (Eq. B12).

$$\int_{x_0}^{x} \frac{1}{x}\,dt = \int_{t_0}^{t} -(k_\mathrm{des} + k_\mathrm{ads})\,dt \tag{B11}$$

$$x = x_0 e^{-(k_\mathrm{des} + k_\mathrm{ads})t} \tag{B12}$$

The equilibration time $\tau_\mathrm{eq}$ can therefore be considered the $e$-folding time required for the displacement $x$ to decrease to $1/e$ of its initial value (Eq. B13).

$$\tau_\mathrm{eq} \approx \frac{1}{k_\mathrm{des} + k_\mathrm{ads}} = \frac{1}{Ae^{-E_\mathrm{A}/RT} + \alpha_\mathrm{s,0}d_\mathrm{p}^2\pi N_\mathrm{p}\omega/4} \tag{B13}$$





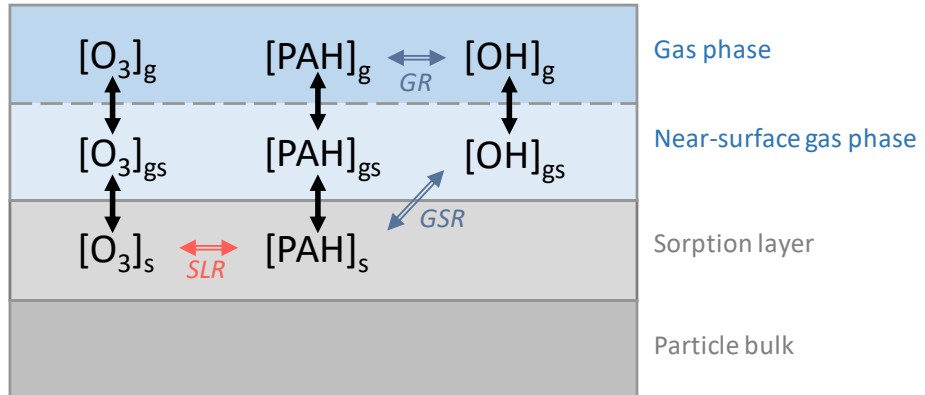

**Figure A1.** The model used in this study consists of three layers: gas-phase, near-surface gas-phase and sorption layer. The surface-layer reaction (SLR) with $O_3$, as well as the gas-surface reactions (GSR) and gas-phase reactions (GR) with OH are modeled alongside mass transport and partitioning.

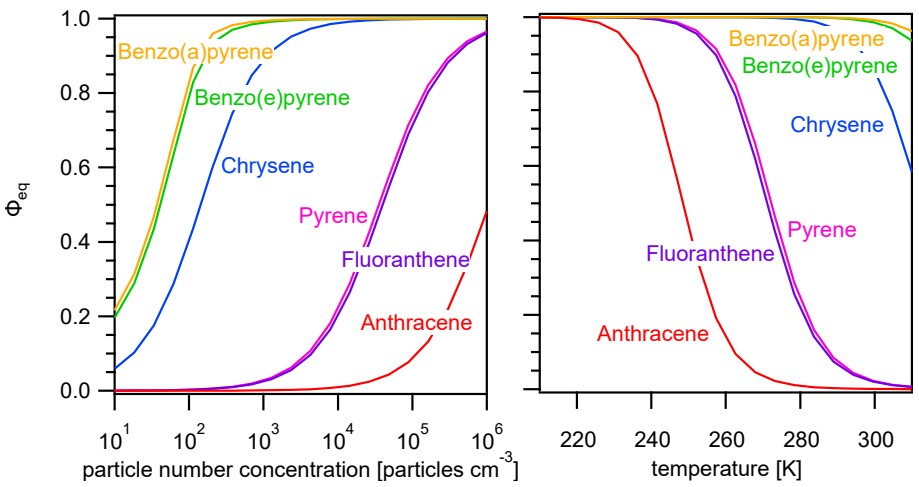

**Figure A2.** Equilibrium particulate fraction $\Phi_{\mathrm{eq}}$ of selected PAHs as a function of a) particle number concentration ($T = 298$ K) and b) temperature ($N_{\mathrm{p}} = 1 \times 10^3$) for particles of size $d_{\mathrm{p}} = 50$ nm.





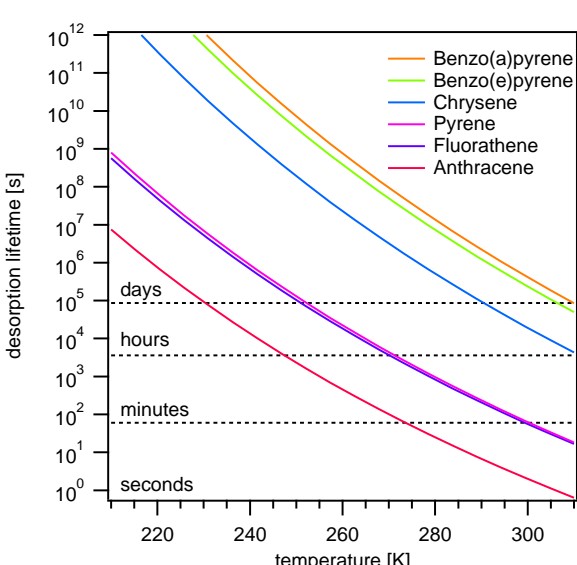

**Figure A3.** Desorption lifetime ($\tau_{\mathrm{des}} = 1/k_{\mathrm{des}}$) of selected PAHs as a function of temperature.



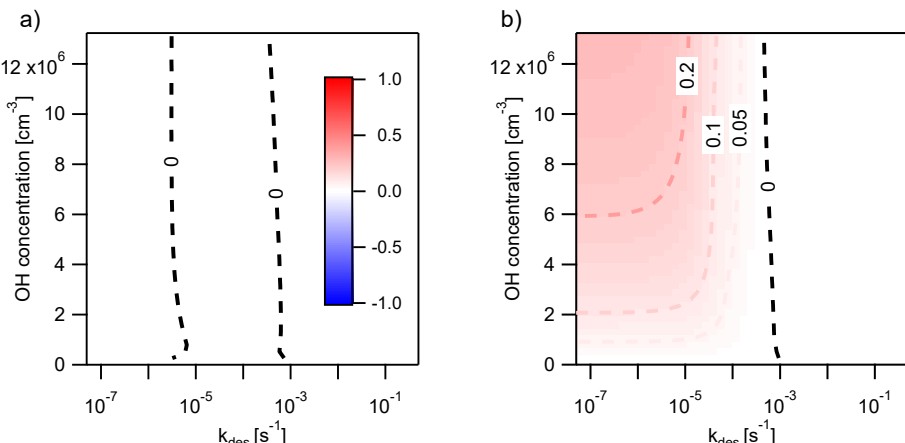

**Figure A4.** Discrepancy, determined with the error metric $E_{\text{loss}}$, between the the explicit-coupled solution EC and the operator-splitting solution IE with a time step of $\Delta t = 30$ min for the reaction of OH (1 ppt) with a) both gas- and particle-phase pyrene and b) only gas-phase pyrene. Regions where EC overestimates the loss of pyrene are indicated in red ($0 < E_{\text{loss}} \leq 1$) and underestimated in blue ($-1 \geq E_{\text{loss}} < 0$). The conditions are 280 K and $1 \times 10^3$ particles cm$^{-3}$.

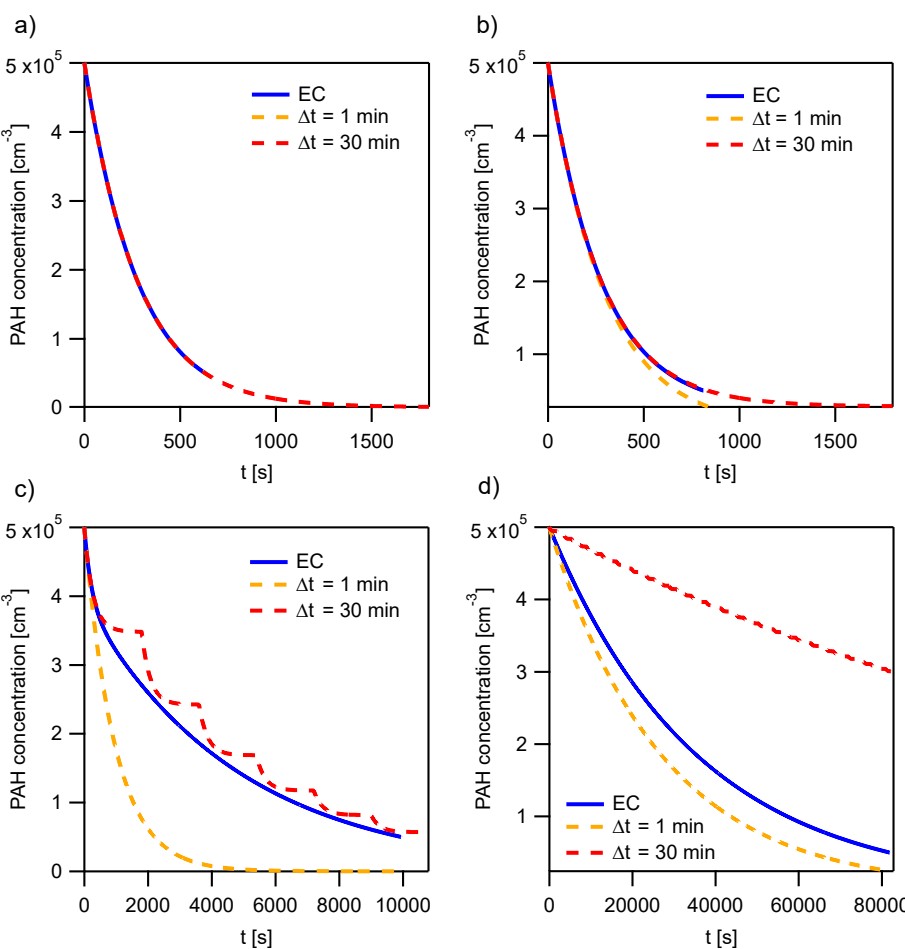

**Figure A5.** The explicit-coupled (EC) solution and instantaneous equilibration (IE) solutions with time steps $\Delta t = 1$ and $30$ min at specific points for the evaluation of the error metric $E_{\text{loss}}$ (Fig. 8a,b). These points represent a horizontal cross section of Fig. 8 at $[O_3] = 2.5 \times 10^{12}$ cm$^{-3}$ and the following $k_{\text{des}}$: a) $10^{-6.3}$ b) $10^{-4.7}$ c) $10^{-3.1}$ and d) $10^{-1.5}$ s$^{-1}$.

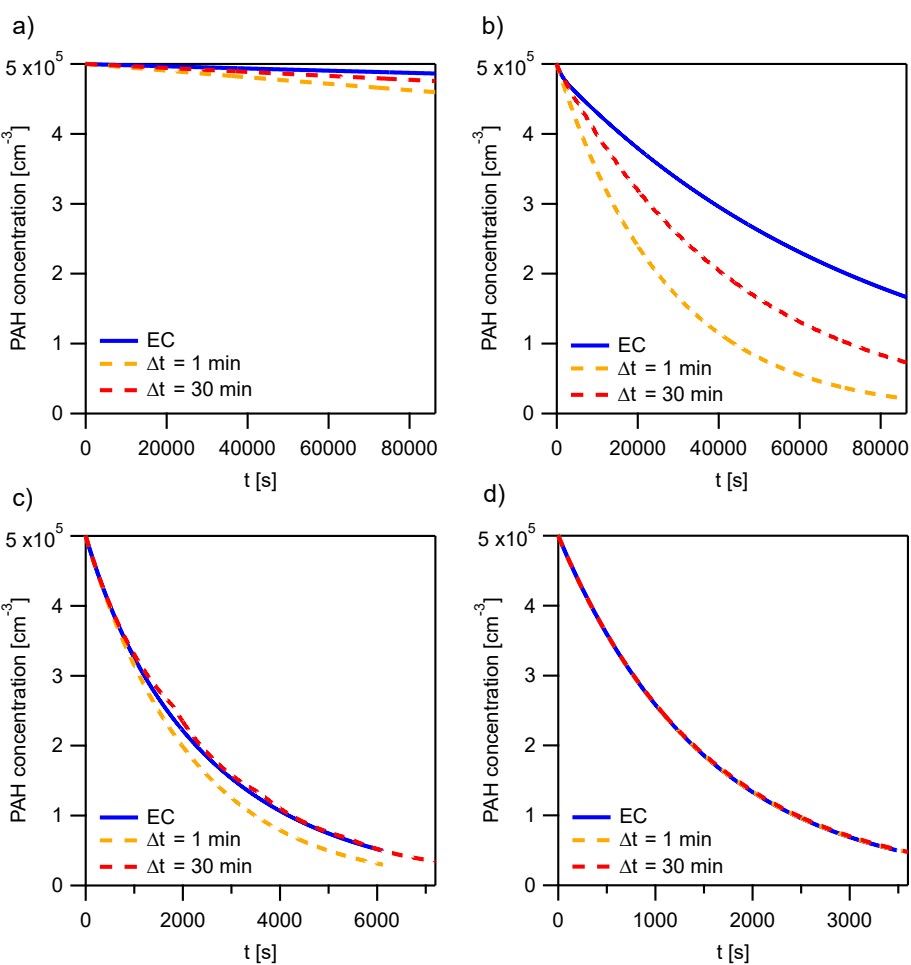

**Figure A6.** The explicit-coupled (EC) solution and instantaneous equilibration (IE) solutions with time steps $\Delta t = 1$ and 30 min at specific points for the evaluation of the error metric $E_{\text{loss}}$ (Fig. A4b). These points represent a horizontal cross section of Fig. A4b at $[OH] = 1 \times 10^7$ cm$^{-3}$ and the following $k_{\text{des}}$: a) $10^{-6.3}$ b) $10^{-4.7}$ c) $10^{-3.1}$ and d) $10^{-1.5}$ s$^{-1}$.


**Table A1.** Mass-transport parameters to determine the desorption rate coefficients of PAHs and $O_3$ ($k_{\text{des}} = Ae^{-E_A/RT}$): Arrhenius parameter $A$ and activation energy of desorption $E_A$ (Guilloteau et al., 2008, 2010).

| Species | $A\,/\,\text{s}^{-1}$ | $E_a\,/\,\text{kJ mol}^{-1}$ |
| --- | --- | --- |
| Anthracene | $1.1 \times 10^{15}$ | 88.1 |
| Fluoranthene | $0.4 \times 10^{15}$ | 93.9 |
| Pyrene | $0.6 \times 10^{15}$ | 95.2 |
| Chrysene | $5.3 \times 10^{15}$ | 114.9 |
| Benzo(e)pyrene | $3.2 \times 10^{15}$ | 119.9 |
| Benzo(a)pyrene | $3.9 \times 10^{15}$ | 121.8 |
| $O_3$ | $1.0 \times 10^{14}$ | 80.0 |

**Table A2.** Mass-transport parameters for adsorption of PAHs and $O_3$: molar mass $M$, surface accommodation coefficient on an adsorbate-free substrate $\alpha_{s,0}$ (Julin et al., 2014) and molecular cross section $\sigma$. The gas-phase diffusion coefficient $D_g$ used for PAHs is $0.06\ \text{cm}^2\text{s}^{-1}$ (Siddiqi et al., 2009) and for $O_3$ is $0.14\ \text{cm}^2\text{s}^{-1}$ (Massman, 1998).

| Species | $M\,/\,\text{g mol}^{-1}$ | $\alpha_{s,0}$ | $\sigma\,/\,\text{cm}^2$ |
| --- | --- | --- | --- |
| Anthracene | 178 | 1 | $6\times10^{-15}$ |
| Fluoranthene | 202 | 1 | $8\times10^{-15}$ |
| Pyrene | 202 | 1 | $8\times10^{-15}$ |
| Chrysene | 228 | 1 | $8\times10^{-15}$ |
| Benzo(e)pyrene | 252 | 1 | $10\times10^{-15}$ |
| Benzo(a)pyrene | 252 | 1 | $10\times10^{-15}$ |
| $O_3$ | 48 | 0.001 | $1.7\times10^{-15}$ |



**Table A3.** Symbols, definitions and units.

| Symbol | Description | Units |
|:---:|:---:|:---|
| $[\text{PAH}]_\text{g}$ | gas phase concentration of PAH | $\text{cm}^{-3}$ |
| $[\text{PAH}]_\text{gs}$ | near-surface gas phase concentration of PAH | $\text{cm}^{-3}$ |
| $[\text{PAH}]_\text{s}$ | surface concentration of PAH | $\text{cm}^{-2}$ |
| $[\text{PAH}]_\text{p}$ | total particle phase concentration of PAH | $\text{cm}^{-3}$ |
| $\Phi$ | particulate fraction | - |
| $\Phi_\text{eq}$ | equilibrium particulate fraction | - |
| $\Phi_\text{qs}$ | quasi-steady-state particulate fraction | - |
| $\Phi_0$ | initial particulate fraction | - |
| $J_\text{diff}$ | gas diffusion flux | $\text{s}^{-1}$ |
| $J_\text{coll}$ | collision flux | $\text{cm}^{-2}\,\text{s}^{-1}$ |
| $J_\text{ads}$ | adsorption flux | $\text{cm}^{-2}\,\text{s}^{-1}$ |
| $J_\text{des}$ | desorption flux | $\text{cm}^{-2}\,\text{s}^{-1}$ |
| $V_\text{gs}$ | volume of near-surface gas phase | $\text{cm}^3$ |
| $L_\text{g}$ | loss rate from gas phase | $\text{cm}^{-3}\,\text{s}^{-1}$ |
| $L_\text{s}$ | loss rate from particle phase | $\text{cm}^{-2}\,\text{s}^{-1}$ |
| $N_\text{p}$ | particle number concentration | $\text{particles}\,\text{cm}^{-3}$ |
| $d_\text{p}$ | particle diameter | cm |
| $t$ | time | s |
| $\tau_\text{eq}$ | equilibration timescale | s |
| $\tau_\text{des}$ | desorption lifetime | s |
| $\tau_\text{ads}$ | adsorption timescale | s |
| $D_\text{g}$ | gas-phase diffusion coefficient | $\text{cm}^2\,\text{s}^{-1}$ |
| $R$ | gas constant | $\text{J}\,\text{K}^{-1}\,\text{mol}^{-1}$ |
| $\omega$ | mean thermal velocity | $\text{cm}\,\text{s}^{-1}$ |
| $M$ | molar mass | $\text{kg}\,\text{mol}^{-1}$ |
| $\sigma$ | molecular cross section | $\text{cm}^2$ |
| $T$ | temperature | K |
| $E_\text{A}$ | activation energy of desorption | $\text{J}\,\text{mol}^{-1}$ |
| $k_\text{des}$ | desorption rate coefficient | $\text{s}^{-1}$ |
| $\alpha_\text{s,0}$ | surface accommodation coefficient on an adsorbate-free substrate | - |



**Table A4.** Symbols, definitions and units.

| Symbol | Description | Units |
|--------|-------------|-------|
| EC | explicit-coupled solution | - |
| IE | instantaneous-equilibration solution | - |
| $[\mathrm{PAH}](t)$ | PAH concentration at time $t$ | $\mathrm{cm}^{-3}\,\mathrm{s}^{-1}$ |
| $[\mathrm{PAH}](0)$ | PAH concentration at time $t = 0$ | $\mathrm{cm}^{-3}\,\mathrm{s}^{-1}$ |
| $\Delta[\mathrm{PAH}](\mathrm{t})$ | total PAH loss up to time $t$ | $\mathrm{cm}^{-3}\,\mathrm{s}^{-1}$ |
| $\Delta t$ | operator-splitting time step | s |
| $\Delta t_{\mathrm{opt}}$ | optimal operator-splitting time step | s |
| $t_{0\%}$ | time at which simulation starts | s |
| $t_{90\%}$ | time at which 90 % of PAH is lost | s |





*Author contributions.* JW, UP, MS and TB designed research. JW, MS and TB wrote the model code and designed model calculations. JW and TB analyzed model results. JW and TB wrote the manuscript with contributions from all co-authors.

*Competing interests.* The authors declare that they have no conflict of interest.

*Acknowledgements.* This study was supported by the Max Planck Society (MPG). MS acknowledges funding by the U.S. National Science Foundation (AGS-1654104). The authors thank Gerhard Lammel for helpful discussions.



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
