# Peer review of "Non-equilibrium interplay between gas-particle partitioning and multiphase chemical reactions of semi-volatile compounds: mechanistic insights and practical implications for atmospheric modeling of PAHs"

_Atmospheric Chemistry and Physics, 2020_

## Referee Comment (RC1) · Anonymous Referee #1 · 24 Nov 2020

This is a very well-written paper. The fundamental concepts of kinetic gas-particle partitioning due to adsorption-desorption processes and surface-chemical reactions are well presented.

The Atmospheric Implications section lists the challenges in terms of application of this conceptual picture to secondary organic aerosols (SOA). I agree that such a simple picture is not directly applicable to SOA where absorptive partitioning, particle-phase

state, and diffusion limitations vary in the atmosphere.

It would be interesting to see how humidity and aerosol water affect the interplay between gas-particle partitioning and chemical reactions in addition to temperature for SOA in future studies.

I recommend acceptance of this paper.
* * *

---

## Referee Comment (RC2) · Anonymous Referee #2 · 9 Dec 2020

General comments: This paper summarizes a computational study investigating the effects of chemical losses (reactions with O3 and OH) on gas-particle partitioning of PAHs on soot particles. This study is important because it reiterates the idea that equilibrium partitioning is not an accurate model of the gas-particle distribution of PAHs, but it further demonstrates that a number of factors can affect the non-equilibrium distribution of PAHs between gases and soot particles and how far from equilibrium these impacts perturb the system. I think it should be published after addressing my comments

below, and also an overarching question I was left with: how much do the authors predict these findings will impact the results of current global and regional CTMs? They conclude by saying that the model is too computationally intensive to be deployed, and that another algorithm summarizing the results could be created, but even if that is done, do they anticipate a large change in results?

Specific comments: It would be useful to add a short section at the end of paragraph 2 in the Introduction that explains why gas-P partitioning is important assessing human health hazards – a short description of the differences between gas-phase and particle-phase deposition behavior and also uptake into lungs.

Line 62: I suggest explicit definitions of SOA and "particles" or "aerosol particles" to aid readers.

Line 76: Does PRA stand for something? If so, would help to define the abbreviation.

Line 78: Quantification of equilibrium processes for six PAHs is mentioned, but this is a bit confusing because the abstract only notes using pyrene as a case study. I suggest clarification in both the abstract and the end of the intro on which parts of the study employ six PAHs and where pyrene is used as a model PAH.

Line 97 through 106: It would be helpful to put the diffusion flux terms on Fig. A1 and to very briefly explain their direction in the main text—it is not clear without reading the appendix that the gas diffusion flux is between the gas phase and the near-surface gas phase. Another thing that kind of confuses me is the representation of the gas phase a finite layer in Fig. A1. It made me wonder why equation 1 didn't contain a term for the influx of gas-phase particles, because I was thinking of it as something separate (as part of the particle) from the infinite bulk gas phase of the atmosphere. Is there a way to graphically represent it as a layer that extends into space on the top side in A1? Finally, the term LS is used in equation 3, but the term LP is defined in the text (line 104).

Line 113: Why the choice to use the same O3 rate coefficient for all PAHs observed?

Line 122: Same question for the rate coefficient for PAHs with OH.

Since the abstract states that the equilibration timescale depends on factors including temperature (and the influence of temperature is mentioned several times throughout the introduction), it is surprising that the temperature dependence of kinetic rate coefficients is not taken into account.

Section 3.1 of the results and Fig. 2 in particular nicely outline the approach of the paper, and though this was hinted at in the intro, I feel that it would be helpful to have something a little more detailed like this earlier on, like in the intro. I couldn't really picture the goals/approach well until reaching this point in the article.

Line 180: There seems to be an entire section missing on PAH molecular structure!

Lines 204-205: The description of the shading in Figure 3 is backwards.

Throughout: the concentration unit of cm-3 is kind of confusing to me. At the beginning of the results, it is explained that PAH concentration is kept at 5 x 105 cm-3. Is that molecules per cm3? And if so, why is not stated? [Also, as an aside, this seems like a detail that should be placed in the methods rather than the results.]

Figure 8: Delta t = 1 min and Delta t = 30 min. should be overlayed on plots a) and b) respectively.

Also Figure 8: It would be interesting for the authors to highlight the area on these plots where they feel most CTMs reside... are desorption kinetics too fast? What about O3 concentrations? If modeled, these usually vary based on simulated chemistry. Also, do most models even include O3 oxidation? It seems to me that many only consider OH. Is there an opportunity here to compare the relative importance of OH and O3 oxidation as a whole to PAH lifetimes?

Technical corrections: Line 70: Replace "accounting" with "account".

---

## Referee Comment (RC3) · Anonymous Referee #3 · 14 Dec 2020

This work described and explored the dynamic and non-equilibrium interplay of gas-particle partitioning and chemical losses of PAHs on soot particles by using a kinetic model. This is a nice work, and should be published in the ACP. I have two major comments. (1) The influence from the absorptive partitioning should be discussed in more detail since in many cases the absorption is more important than the adsorption in gas-particle partitioning for PAHs in the atmosphere; and (2) It would be helpful that the results obtained from their theoretical study can be evaluated by monitoring data.

[Figure]

Technical corrections: Line 7: "to span seconds to hours" -> "to span from seconds to hours" Line 58: "SOA" -> "secondary organic aerosol (SOA)" Line 62: "secondary organic aerosol (SOA)" -> "SOA" Line 95: The authors should explain why cm3 can be used as the unit of concentrations of PAH.

———————————————

---

## Author Comment (AC1) · 29 Jan 2021

We would like to thank the editor and referees for their contributions to the review process. In the following, we will provide point-by-point answer to the reviewer's comments and suggestions. The reviewer's comments are given in **black**, our responses in **blue**, and updates to the manuscript in **orange**.

**Response to Anonymous Referee #1**

This is a very well-written paper. The fundamental concepts of kinetic gas-particle partitioning due to adsorption-desorption processes and surface-chemical reactions are well presented. The Atmospheric Implications section lists the challenges in terms of application of this conceptual picture to secondary organic aerosols (SOA). I agree that such a simple picture is not directly applicable to SOA where absorptive partitioning, particle-phase state, and diffusion limitations vary in the atmosphere. It would be interesting to see how humidity and aerosol water affect the interplay between gas-particle partitioning and chemical reactions in addition to temperature for SOA in future studies. I recommend acceptance of this paper.

Response: We would like to thank Anonymous Referee #1 for their positive review and comments. We are also very interested to see how humidity and water content affect the kinetics of partitioning and hope to follow up on this in a future study.

**Response to Anonymous Referee #2**

General comments: This paper summarizes a computational study investigating the effects of chemical losses (reactions with O3 and OH) on gas-particle partitioning of PAHs on soot particles. This study is important because it reiterates the idea that equilibrium partitioning is not an accurate model of the gas-particle distribution of PAHs, but it further demonstrates that a number of factors can affect the non-equilibrium distribution of PAHs between gases and soot particles and how far from equilibrium these impacts perturb the system. I think it should be published after addressing my comments below, and also an overarching question I was left with: how much do the authors predict these findings will impact the results of current global and regional CTMs? They conclude by saying that the model is too computationally intensive to be deployed, and that another algorithm summarizing the results could be created, but even if that is done, do they anticipate a large change in results?

Response: We would like to thank Anonymous Referee #2 for their detailed review and useful comments. The impact of our results on current global and regional CTMs is indeed an interesting point. In order to fully quantify the error introduced by the instantaneous equilibration assumption, it would be necessary to implement a non-equilibrium partitioning scheme directly into a CTM. Furthermore, temperature, PAH volatility, particle number concentration, oxidant concentration, and operator splitting time step influence the magnitude of differences between both calculation methods, as shown in Fig. 8 of our manuscript. While in some modelling scenarios there will be no difference, there will be large discrepancies in others. The metric $E_{loss}$ in Fig. 8, however, can provide a possible range. For instance, if $E_{loss}$ = -0.5, a model using the instantaneous equilibration (IE) assumption would be underestimating the loss of PAH by 34 %. Likewise, if $E_{loss}$ = 0.1, a model using the instantaneous equilibration assumption would be overestimating the loss of PAH by 23 %. We have added these figures to the manuscript to give the reader a general idea about the expectable deviations.

Line 388: *"To estimate errors for global models, it is also informative to present the discrepancy between the EC and IE approaches as a percentage difference. In Fig. 8 for instance, when $E_{loss}$ = -0.5, a model using the IE assumption underestimates the loss of PAH by 34 % compared to the EC solution*

*(Fig. 8a). Likewise, if E$_{loss}$ = -0.1, a model using the IE assumption overestimates the loss of PAH by 23 %.  In order to fully quantify the error introduced by the instantaneous equilibration assumption, it would be necessary to implement a non-equilibrium partitioning scheme directly into a CTM."*

Although these values offer a guideline to the magnitude to error that could be expected, we reiterate that this needs to be tested by implementing a non-equilibrium partitioning scheme for PAHs. A key aim of this manuscript is to lay the foundations for non-equilibrium effects of PAHs to be systematically included in CTMs.

Alongside PAH concentration, CTMs are often evaluated by comparing the predicted PAH gas-particle partitioning to observational data. The particulate fraction and partitioning coefficient both metrics depend on the relative concentrations between gas-phase and particle-phase PAHs. Compared with absolute PAH concentration, partitioning metrics could be equally sensitive, if not more, to the effects of a non-equilibrium partitioning scheme. This is also now mentioned in the manuscript:

Line 403: *"It should be noted that alongside the gas-phase and particle-phase concentrations of PAHs, CTMs are often evaluated by comparing the predicted particulate fraction and partitioning coefficient to observational data. Both of these metrics depend on the relative concentrations of gas-phase and particle-phase PAHs. Therefore, due to error propagation, the particulate fraction and partitioning coefficient may be more sensitive than absolute concentration to the effects of a non-equilibrium partitioning scheme."*

Specific comments: It would be useful to add a short section at the end of paragraph 2 in the Introduction that explains why gas-P partitioning is important assessing human health hazards – a short description of the differences between gas-phase and particle-phase deposition behavior and also uptake into lungs.

Response: We have now included details on the direct effect of partitioning on health: characteristics of deposition and bioavailability depend on where a compound predominantly resides, the gas phase or the particle phase.

Line 33: *"Moreover, after inhalation, the distribution of a semi-volatile compound between the gas phase and the particle phase can determine its bioaccessibility (Liu et al, 2017; Wei et al., 2020). While gas-phase PAHs can directly partition into the epithelial lining fluid of the lung, particle-phase PAHs first have to dissolve from a matrix and may hence be less bioaccessible (Lammel et al., 2020)."*

Line 62: I suggest explicit definitions of SOA and "particles" or "aerosol particles" to aid readers.

Response: We have now provided explicit definitions of aerosol particles and SOA.

Line 26: "*PAHs are semi-volatile compounds that may exist in the gas phase, adsorbed on the surface of aerosol particles, or absorbed into the bulk of particles. The transport and distribution between these phases is referred to as gas-particle partitioning.*"

Now reads:

"*PAHs are semi-volatile compounds that may exist in the gas phase, adsorbed on the surface, or absorbed into the bulk of aerosol particles. As atmospheric aerosols, we describe the suspension of nano- to micrometer sized particles in outside air. Typical atmospheric aerosol particles include sea salt, mineral dust, sulfate, and organic particles* (Pöschl, 2005). *Two key types of organic particles include soot, formed during fossil-fuel combustion, and secondary organic aerosols (SOA), formed from condensation of organic vapors in the atmosphere. The mass transfer and distribution of PAHs between the gas phase and particle phase is referred to as gas-particle partitioning.*"

Line 76: Does PRA stand for something? If so, would help to define the abbreviation.

Response: The 'PRA' acronym refers to the framework developed by three authors: Pöschl, Rudich, and Ammann, in the reference Pöschl et al. 2007. We agree it is helpful to clarify this and have now defined the abbreviation on line 84 of the manuscript.

Line 78: Quantification of equilibrium processes for six PAHs is mentioned, but this is a bit confusing because the abstract only notes using pyrene as a case study. I suggest clarification in both the abstract and the end of the intro on which parts of the study employ six PAHs and where pyrene is used as a model PAH.

Response: We now signal more clearly in the abstract and the introduction the cases for which six PAHs were used and the cases for which only pyrene was used.

Line 14: *"Conversely, perturbations are smaller for reaction with the OH radical, which reacts with PAHs on both the surface of particles and in the gas phase.*

Now reads:

*"Conversely, perturbations are smaller for reaction with the OH radical, which reacts with pyrene on both the surface of particles and in the gas phase."*

Line 82: *"In this study, we examine the timescales of gas-particle partitioning and chemical loss of PAHs with a kinetic model in which both processes are explicitly coupled."*

Now reads:

*"In this study, we use a kinetic model to 1) examine the timescales of gas-particle partitioning for six PAHs and 2) investigate the chemical loss of PAHs by explicitly coupling the partitioning and oxidation chemistry of the PAH pyrene."*

Line 87: *"We illustrate how the combination of slow partitioning and chemical loss of PAHs can perturb the particulate fraction from equilibrium (Sect. 3.3.2) and alter chemical lifetime (Sect. 3.3.1)."*

Now reads:

*"We illustrate how the combination of slow partitioning and chemical loss of PAHs can perturb the particulate fraction from equilibrium (Sect. 3.3.2) and alter chemical lifetime (Sect. 3.3.1) in the example of the PAH pyrene."*

In Section 3.4.2 we now indicate more clearly that these results are for a range of PAHs with different desorption rates, not just pyrene.

Line 354: *"In this section, this discrepancy is explored as a function of desorption rate and is therefore characteristic of a range of PAHs."*

Line 97 through 106: It would be helpful to put the diffusion flux terms on Fig. A1 and to very briefly explain their direction in the main text. It is not clear without reading the appendix that the gas diffusion flux is between the gas phase and the near-surface gas phase.

Response: We have added the diffusion flux terms to Fig. A1. The caption is updated accordingly and a description of the direction of fluxes given in the main text.

Fig. A1 Caption: *"The surface-layer reaction (SLR) with $O_3$, as well as the gas-surface reactions (GSR) and gas-phase reactions (GR) with OH are modeled alongside mass transport and partitioning."*

Now reads:

*"The surface-layer reaction (SLR) with $O_3$, as well as the gas-surface reactions (GSR) and gas-phase reactions (GR) with OH are modeled alongside mass transport and partitioning. $J_{diff}$, $J_{des}$, and $J_{ads}$ represent the gas diffusion flux between the gas phase and the near-surface gas phase, the desorption flux from the particle surface to the near-surface gas phase, and the adsorption flux from the near-surface gas phase to the particle surface, respectively. Note that for the gas diffusion flux of OH, gas diffusion was implemented through a correction factor $C_{g,OH}$."*

Another thing that kind of confuses me is the representation of the gas phase a finite layer in Fig. A1. It made me wonder why equation 1 didn't contain a term for the influx of gas-phase particles, because I was thinking of it as something separate (as part of the particle) from the infinite bulk gas phase of the atmosphere. Is there a way to graphically represent it as a layer that extends into space on the top side in A1?

We have adjusted the shading in Fig. A1 so that the gas layer extends into space.

Original Fig. A1:

[Figure]

*Mass transport and partitioning*
*Chem. reaction with $O_3$*
*Chem. reaction with OH*

Updated Fig. A1:

[Figure]

*Mass transport and partitioning*
*Chem. reaction with $O_3$*
*Chem. reaction with OH*

Finally, the term LS is used in equation 3, but the term LP is defined in the text (line104).

Thank you for spotting the inconsistent use of $L_s$ vs $L_p$ to describe the loss rate of pyrene from the particle phase. $L_p$ was an artifact from a previous draft of the manuscript. The symbol $L_s$ for the loss in the particle phase is now consistent throughout the manuscript.

Line 113: Why the choice to use the same O3 rate coefficient for all PAHs observed? Line 122: Same question for the rate coefficient for PAHs with OH.

Response: The $O_3$ and OH rate coefficients are used only for pyrene and are not directly applied to other PAHs. We believe the confusion originates, in part, for the reasons mentioned in the previous comment. To help further clarify this point, we have replaced 'PAH' with pyrene throughout section 2.2.

An exception occurs in Section 3.4.2 "Deviation from explicit-coupled (EC) approach". The reaction rate coefficients of pyrene are used as "best guesses" for generic PAHs, represented by varying desorption rates. We acknowledge that in reality rate coefficients may differ, but decided to exclude these secondary effects in order to explore the effects of non-equilibrium partitioning on chemical loss. To make this clear to the reader, we add:

Line 356: "The reaction rate coefficients of pyrene are used as best guess for generic PAHs."

Since the abstract states that the equilibration timescale depends on factors including temperature (and the influence of temperature is mentioned several times throughout the introduction), it is surprising that the temperature dependence of kinetic rate coefficients is not taken into account.

Response: For the surface reaction between $O_3$ and pyrene, we expect the main driver of sensitivity in the model with regard to temperature to be the desorption lifetime. The temperature-dependent desorption rates of both reactants are included in the model, making the gas-particle partitioning of PAHs and $O_3$ and hence the chemical loss temperature dependent. It is possible that the accommodation coefficient and the surface layer reaction rate coefficient for the reaction between pyrene and $O_3$ are temperature dependent. Unfortunately, to the best of our knowledge, the temperature dependence of these parameters has not been studied.

In the case of the reaction between OH and pyrene, we do not expect a significant dependence on temperature, given the high reactivity of OH. In accordance, the uptake coefficient of OH on a pyrene surface has been found to increase only slightly with temperature between 218 to 298 K in experiments (Liu et al., 2012). Likewise, in an experimental study of six PAHs, the temperature dependence of the rate coefficient was judged to be 'slight to nonexistent' (Brubaker and Hites, 1998). We have included this in the manuscript.

Line Z: "*We do not consider temperature dependence of chemical rate coefficients in this model.*"

Now reads:

Line 141: *"The temperature dependence of the gas-phase OH reaction of PAHs has been found experimentally to be 'slight to nonexistent' (Brubaker and Hites, 1998). Likewise, the reaction probability of OH on a pyrene surface has been found to exhibit only a slight temperature dependence (Liu et al., 2012). We therefore do not include temperature dependence of chemical rate coefficients in this model."*

Line 129: *"The desorption rate coefficients of both pyrene and O₃, which are temperature dependent and explicitly included in the model, are expected to be the main driver of sensitivity in the model with regards to temperature. It should be noted that the accommodation coefficient and surface layer reaction rate coefficient may also exhibit temperature dependence, but without further quantitative parameters, cannot be included in the model."*

Section 3.1 of the results and Fig. 2 in particular nicely outline the approach of the paper, and though this was hinted at in the intro, I feel that it would be helpful to have something a little more detailed like this earlier on, like in the intro. I couldn't really picture the goals/approach well until reaching this point in the article.

We particularly value this comment and we carefully considered how Fig. 2 could be introduced earlier on. Unfortunately, we couldn't find a suitable location in the introduction to place this figure and text, and didn't feel it was appropriate in the methods. We did however introduce some additional text in the introduction to help better describe the key effect that Fig. 2 is illustrating.

Line 88: "We detail how a dominant loss of pyrene from the particle phase may decrease the particulate fraction. Likewise, in the case of dominant loss of pyrene from the gas phase, the particulate fraction would increase. Compared to instantaneous partitioning, which would conserve equilibrium particulate fractions, chemical lifetime may be affected through depletion of pyrene in the more reactive phase."

Line 180: There seems to be an entire section missing on PAH molecular structure!

Response: Thank you for identifying this mistake. This subtitle was an artifact from an older version of the manuscript and should have been deleted. The information originally contained within this section is already integrated elsewhere in the text. We have now removed this subtitle and renumbered the remaining subsections in section 3.2 accordingly.

Lines 204-205: The description of the shading in Figure 3 is backwards.

Response: Many thanks for spotting this error. It has now been fixed.

Throughout: the concentration unit of cm-3 is kind of confusing to me. At the beginning of the results, it is explained that PAH concentration is kept at 5 x 105 cm-3. Is that molecules per cm3? And if so, why is not stated?

Response: The units of PAH concentration 'cm$^{-3}$' and 'cm$^{-2}$' refer to the number concentration of molecules per cm$^3$ and cm$^2$, respectively, consistent with the Pöschl-Rudich-Ammann (PRA) convention (Pöschl et al., 2007) .

Line 106: *"The differential equations in Eqs. 1-3 describe the time evolution of [PAH]$_g$ (cm$^{-3}$), [PAH]$_{gs}$ (cm$^{-3}$) and [PAH]$_s$ (cm$^{-2}$), which are the concentrations of PAH in the gas phase, near-surface gas phase and on the surface of aerosol particles, respectively."*

Now reads:

*"The differential equations in Eqs. 1-3 describe the time evolution of [PAH]$_g$ (cm$^{-3}$), [PAH]$_{gs}$ (cm$^{-3}$) and [PAH]$_s$ (cm$^{-2}$), which are the number concentrations (i.e. a unitless count of molecules per unit volume or unit area) of PAH in the gas phase, near-surface gas phase and on the surface of aerosol particles, respectively."*

[Also, as an aside, this seems like a detail that should be placed in the methods rather than the results.]

We felt it was useful to report the initial PAH concentration in the results section, primarily because it helps the reader to interpret the results in Fig. 5 without having to refer to the methods again.

Figure 8: Delta t = 1 min and Delta t = 30 min. should be overlayed on plots a) and b)respectively.

Response: That is a great suggestion and helps to make Figure 8 more comprehensible. We have added the labels 'Δt = 1 min' and 'Δt = 30 min' to Figure 8a and b, respectively.

Also Figure 8: It would be interesting for the authors to highlight the area on these plots where they feel most CTMs reside...are desorption kinetics too fast? What about O3concentrations? If modeled, these usually vary based on simulated chemistry. Also, do most models even include O3 oxidation? It seems to me that many only consider OH. Is there an opportunity here to compare the relative importance of OH and O3 oxidation as a whole to PAH lifetimes?

Response:

The aim of Fig. 8 was to explore under which conditions the error resulting from assuming instantaneous equilibrium was largest. The results in Fig. 8 are only for a single temperature and particle number concentration, therefore in order to make more general judgements, it would be necessary to systematically evaluate a range of atmospheric conditions.

For Fig. 8, the range of $O_3$ concentrations (between 0 and 120ppb) was chosen to capture typical atmospheric conditions from a relatively clean background with 10 ppb $O_3$ (Vingarzan, 2004) to a highly-polluted area with 100 ppb $O_3$ (Wang et al., 2017). We agree that the concentration of $O_3$ will vary within models depending on the local chemistry in the area being modelled. To help guide the reader in Fig. 8, we have added an arrow to indicate typical $O_3$ concentrations from a polluted atmosphere to a pristine atmosphere. We have also highlighted desorption rates for three PAHs (Chrysene, Pyrene, an Anthracene) at 280 K.

Original Fig. 8:

[Figure]

Updated Fig. 8:

[Figure]

Added to Fig. 8 Caption: "Desorption rates for the PAHs chrysene, pyrene and anthracene at 280 K are indicated on the top axis."

Most CTMs of PAHs do include the reaction of $O_3$ with particle-phase PAHs (Sehili and Lammel, 2007; Friedman et al., 2014; Mu et al., 2018; Octaviani et al., 2019). Though, there are some examples of models which instead focus on OH reactivity (Galarneau et al., 2014).

We appreciate the great suggestion to compare the relative importance of OH and $O_3$ for PAH lifetimes. Though we feel this goes beyond the scope of this already extended manuscript and hope to explore this topic in detail in a future study.

Technical corrections: Line 70: Replace "accounting" with "account".

Response: Thanks for spotting this. We have corrected the text.

**Response to Anonymous Referee #3**

This work described and explored the dynamic and non-equilibrium interplay of gas-particle partitioning and chemical losses of PAHs on soot particles by using a kinetic model. This is a nice work, and should be published in the ACP.

Response: We thank Anonymous Referee #3 for their useful comments.

I have two major comments. (1) The influence from the absorptive partitioning should be discussed in more detail since in many cases the absorption is more important than the adsorption in gas-particle partitioning for PAHs in the atmosphere;

Response: In this study, we focus our discussion on the partitioning of PAHs onto the surface of soot particles for which adsorptive partitioning is expected to dominate overall partitioning. Absorptive partitioning is discussed in the introduction (p. 3, l. 75-79) and in the Atmospheric Implications section (p. 19-20, l. 399-407). We do recognize the importance of absorptive partitioning and intend to address it in detail with a separate follow-up study. For now, however, we have added some discussion to clarify the expected difference in equilibration timescales for absorptive partitioning and adsorptive partitioning.

Line 422: "Depending on the controlling regime, it might be expected that absorption lengthens the equilibration timescale compared to adsorption on a solid surface."

Now reads:

Line 422: "For absorptive partitioning, the equilibration timescales of PAHs are expected to be even longer than the equilibration timescales for adsorptive partitioning. Alongside the contributions to the equilibration timescale from the adsorption and the desorption controlling regimes, absorptive partitioning is also controlled by the diffusion of PAHs through the bulk of aerosol particles."

(2) It would be helpful that the results obtained from their theoretical study can be evaluated by monitoring data.

Response: We agree with the reviewer and recognize the importance of using monitoring data to validate the model. It would be possible to simulate an air parcel containing PAHs, soot aerosol particles, OH, and $O_3$ with the model in its current form. We would however need monitoring data with a high temporal resolution and not subject to interferences such as emissions from multiple

sources. Ideally, emissions of PAHs would be arriving at the monitoring station from a single point source and atmospheric variables such as temperature, particle number concentration, and oxidant concentration would also have to be well constrained.

While technically possible, we feel this would be beyond the scope of this manuscript, which aims to lay down the conceptual and mathematical foundations of non-equilibrium partitioning. In a follow up study, we aim to establish how the magnitude of the deviation in particulate fraction between observed values and the predictions of equilibrium partitioning depends on the concentrations of OH, $O_3$, and other perturbing variables. This could be achieved by combining air parcel models with existing observational data. We have now mentioned this possibility in the Atmospheric Implications section.

Line 439: "Using existing observational datasets, it may be possible to establish how the size of the deviation in particulate fraction (between observed values and the predictions of equilibrium partitioning models) depend on the concentrations of OH, $O_3$, $NO_3$ and other perturbing variables. This could help identify and compare key perturbing variables in a real-world setting."

Technical corrections: Line 7: "to span seconds to hours" -> "to span from seconds to hours"

We accept the suggested change.

Line 58: "SOA" -> "secondary organic aerosol (SOA)"

We accept the suggested change.

Line 62: "secondary organic aerosol (SOA)" -> "SOA"

We accept the suggested change.

Line 95: The authors should explain why cm3 can be used as the unit of concentrations of PAH.

Response: As detailed in the response to reviewer #2, the units of PAH concentration '$cm^{-3}$' and '$cm^{-2}$' refer to the number concentration of molecules per $cm^3$ and $cm^2$, respectively, consistent with the Pöschl-Rudich-Ammann (PRA) convention (Pöschl et al., 2007) .

Line 106: *"The differential equations in Eqs. 1-3 describe the time evolution of $[PAH]_g$ ($cm^{-3}$), $[PAH]_{gs}$ ($cm^{-3}$) and $[PAH]_s$ ($cm^{-2}$), which are the concentrations of PAH in the gas phase, near-surface gas phase and on the surface of aerosol particles, respectively."*

Now reads:

*"The differential equations in Eqs. 1-3 describe the time evolution of $[PAH]_g$ ($cm^{-3}$), $[PAH]_{gs}$ ($cm^{-3}$) and $[PAH]_s$ ($cm^{-2}$), which are the number concentrations (i.e. a unitless count of molecules per unit volume or unit area) of PAH in the gas phase, near-surface gas phase and on the surface of aerosol particles, respectively."*

**References**

Brubaker, W.W., Hites, R. a., 1998. OH Reaction Kinetics of Polycyclic Aromatic Hydrocarbons and Polychlorinated Dibenzo-p-dioxins and Dibenzofurans. J Phys Chem A 102, 915–921. https://doi.org/10.1021/jp9721199

Friedman, C.L., Pierce, J.R., Selin, N.E., 2014. Assessing the influence of secondary organic versus primary carbonaceous aerosols on long-range atmospheric PAH transport. Environ. Sci. Technol. https://doi.org/10.1021/es405219r

Galarneau, E., Makar, P.A., Zheng, Q., Narayan, J., Zhang, J., Moran, M.D., Bari, M.A., Pathela, S., Chen, A., Chlumsky, R., 2014. PAH concentrations simulated with the AURAMS-PAH chemical

transport model over Canada and the USA. Atmospheric Chem. Phys. 14, 4065–4077. https://doi.org/10.5194/acp-14-4065-2014

Lammel, G., Kitanovski, Z., Kukučka, P., Novák, J., Arangio, A.M., Codling, G.P., Filippi, A., Hovorka, J., Kuta, J., Leoni, C., Příbylová, P., Prokeš, R., Sáňka, O., Shahpoury, P., Tong, H., Wietzoreck, M., 2020. Oxygenated and Nitrated Polycyclic Aromatic Hydrocarbons in Ambient Air—Levels, Phase Partitioning, Mass Size Distributions, and Inhalation Bioaccessibility. Environ. Sci. Technol. 54, 2615–2625. https://doi.org/10.1021/acs.est.9b06820

Liu, Y., Ivanov, A., Zelenov, V., Molina, M., 2012. Temperature dependence of OH uptake by carbonaceous surfaces of atmospheric importance. Russ. J. Phys. Chem. B 6, 327–332.

Mu, Q., Shiraiwa, M., Octaviani, M., Ma, N., Ding, A., Su, H., Lammel, G., Pöschl, U., Cheng, Y., 2018. Temperature effect on phase state and reactivity controls atmospheric multiphase chemistry and transport of PAHs. Sci. Adv. 4. https://doi.org/10.1126/sciadv.aap7314

Octaviani, M., Tost, H., Lammel, G., 2019. Global simulation of semivolatile organic compounds – development and evaluation of the MESSy submodel SVOC (v1.0). Geosci. Model Dev. 12, 3585–3607. https://doi.org/10.5194/gmd-12-3585-2019

Pöschl, U., 2005. Atmospheric Aerosols: Composition, Transformation, Climate and Health Effects. Angew. Chem. Int. Ed. 44, 7520–7540. https://doi.org/10.1002/anie.200501122

Pöschl, U., Rudich, Y., Ammann, M., 2007. Kinetic model framework for aerosol and cloud surface chemistry and gas-particle interactions – Part 1: General equations, parameters, and terminology. Atmospheric Chem. Phys. 7, 5989–6023. https://doi.org/10.5194/acp-7-5989-2007

Sehili, A.M., Lammel, G., 2007. Global fate and distribution of polycyclic aromatic hydrocarbons emitted from Europe and Russia. Atmos. Environ. 41, 8301–8315. https://doi.org/10.1016/j.atmosenv.2007.06.050

Vingarzan, R., 2004. A review of surface ozone background levels and trends. Atmos. Environ. 38, 3431–3442. https://doi.org/10.1016/j.atmosenv.2004.03.030

Wang, T., Xue, L., Brimblecombe, P., Lam, Y.F., Li, L., Zhang, L., 2017. Ozone pollution in China: A review of concentrations, meteorological influences, chemical precursors, and effects. Sci. Total Environ. 575, 1582–1596. https://doi.org/10.1016/j.scitotenv.2016.10.081